# Injecting Distributional Awareness into MLLMs via Reinforcement Learning for Deep Imbalanced Regression

Yao Du[1]   Shanshan Song[1,2]   Xiaomeng Li[*,1,2]

## Abstract

Multimodal large language models (MLLMs) struggle with numerical regression under long-tailed target distributions. Token-level supervised fine-tuning (SFT) and point-wise regression rewards bias learning toward high-density regions, leading to regression-to-the-mean behavior and poor tail performance. We identify the lack of cross-sample relational supervision as a key limitation of existing MLLM training paradigms. To address it, we propose a distribution-aware reinforcement learning framework based on Group Relative Policy Optimization, which introduces batch-level comparison-based supervision via the Concordance Correlation Coefficient-based reward to align predicted and ground-truth distributions in terms of correlation, scale, and mean. The framework is plug-and-play, requiring no architectural modification. Experiments on a unified suite of long-tailed regression benchmarks show consistent improvements over SFT and existing MLLM regression methods, with particularly strong gains in medium- and few-shot regimes.

## 1. Introduction

Real-world regression tasks often exhibit long-tailed target distributions, where a small subset of values dominates the training data while many valid targets are sparsely observed. In multimodal large language models (MLLMs), continuous quantities are generated autoregressively as discrete token sequences and optimized via next-token prediction with cross-entropy. While effective for linguistic generation, this training paradigm is fundamentally misaligned with numerical regression, where targets are continuous (Spithourakis & Riedel, 2018). Serializing continuous values into discrete

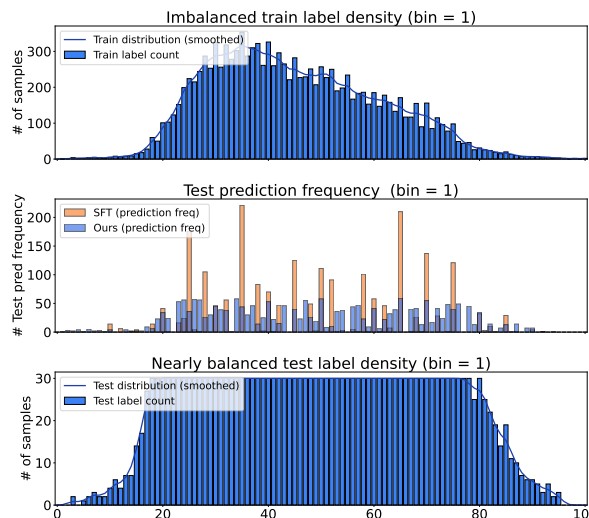

*Figure 1.* SFT exhibits a pronounced regression-to-the-mean effect, with predictions collapsing toward the many-shot region. Our method produces a substantially more balanced prediction distribution and maintains reliable predictions in tail regions.

tokens reduces regression to token-level likelihood maximization with hard one-hot supervision, under which predictions with different numeric errors can incur identical loss as long as they correspond to the same ground-truth token. Consequently, standard supervised fine-tuning (SFT) fails to encode numerical proximity, ordering, or global magnitude, leading to systematic bias in number-related tasks. Under long-tailed supervision, this misalignment further amplifies dominant numeric patterns and provides weak corrective signals for rare targets, resulting in pronounced regression-to-the-mean behavior (Figure 1). Although regression-style objectives are more suitable in principle, integrating them into MLLMs remains nontrivial. Existing approaches rely on architectural modifications (Jiang et al., 2025; Guo et al., 2025b), explicit reasoning procedures (Wang et al., 2025a; Yu et al., 2025), or loss-level adjustments (Wang et al., 2025b), but these strategies either disrupt the unified generative framework, incur substantial inference overhead, or remain confined to local, token-level supervision. Consequently, they fail to capture the global structure and distributional relationships inherent in long-tailed numeric targets. These limitations motivate post-training strategies

[1]The Hong Kong University of Science and Technology, Hong Kong SAR, China [2]Shenzhen Loop Area Institute, Shenzhen, China. Correspondence to: Xiaomeng Li <eexmli@ust.hk>.

*Proceedings of the 43rd International Conference on Machine Learning*, Seoul, South Korea. PMLR 306, 2026. Copyright 2026 by the author(s).

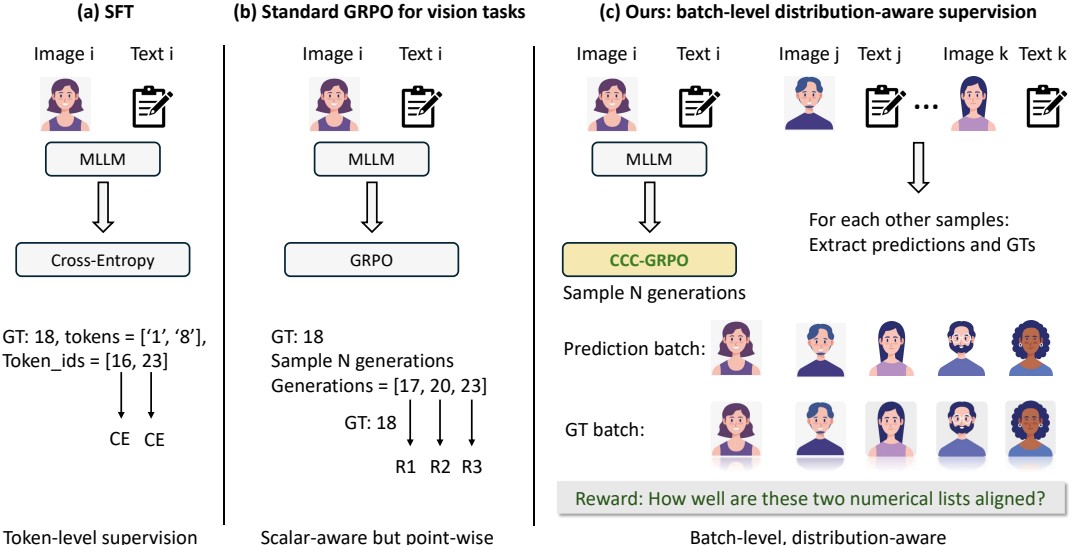

*Figure 2.* Comparison of training paradigms for numerical prediction in MLLMs. **Left:** SFT treats regression as token-level classification. **Middle:** Standard GRPO applies point-wise scalar rewards to each generation. **Right:** CCC-GRPO introduces batch-level, distribution-aware relational supervision.

that operate directly on holistic numerical outputs, without altering model architecture.

Reinforcement fine-tuning has recently emerged as an effective paradigm for training large reasoning models, where supervision is defined over complete generated sequences rather than applied directly at the token level. Representative work such as DeepSeek-R1 (Guo et al., 2025a) shows that RL with verifiable, rule-based rewards can improve generalization and recover key capabilities of proprietary reasoning models like OpenAI o1 (Jaech et al., 2024). This sequence-level formulation is particularly appealing for numerical regression, where targets are continuous and ordered, and token-level SFT fails to capture global numeric structure. Recent studies have applied RL to MLLMs for visual reasoning and perception tasks (Liu et al., 2025b; Shen et al., 2025; Yu et al., 2025). However, existing reward designs remain largely *per-sample* and *local*, relying on simple discriminative signals and neglecting cross-sample relationships and global distributional structure. As a result, the core challenge of deep imbalanced regression—preserving numerical continuity and robustness under long-tailed supervision—remains largely unexplored in current RL-based MLLM frameworks.

Figure 2 illustrates the fundamental differences between existing training paradigms and our proposed approach. Under standard SFT, numerical regression is implicitly cast as token-level classification. Such supervision is inherently insensitive to numerical magnitude and global ordering, and fails to distinguish predictions that are numerically closer but token-wise different. Recent RL approaches like GRPO partially alleviate this issue by operating at the sequence

level. As illustrated in Fig. 2(middle), standard GRPO samples multiple generations and assigns scalar rewards (e.g., MAE variants) to each output (Li et al., 2025). While this makes predictions value-aware, the reward remains *point-wise*: each prediction is evaluated independently against its ground truth. Consequently, the optimization still favors collapsing predictions toward high-density regions, offering limited resistance to long-tailed imbalance. In contrast, our method introduces a *batch-level, distribution-aware* reinforcement learning objective. As shown in Fig. 2(right), for each sampled prediction we construct a relational comparison set that jointly considers the current output and the mean predictions of other samples within the same mini-batch. By computing rewards via the Concordance Correlation Coefficient, our approach explicitly enforces consistency between the predicted and ground-truth distributions in terms of correlation, scale, and mean. This relational supervision penalizes degenerate solutions such as mean collapse, and naturally amplifies the learning signal from under-represented (tail) samples.

Overall, we argue that DIR in MLLMs should be treated as a *distribution-aware learning problem*, rather than a collection of independent point-wise predictions. The key challenge lies in designing appropriate supervision semantics rather than modifying model architectures or optimization algorithms. We propose a principled GRPO framework tailored for DIR. Instead of optimizing per-sample numerical errors in isolation, our method leverages batch-level relational structure and optimizes correlation-based rewards that explicitly account for agreement in both scale and distribution. This design naturally counteract the dominance of densely populated target regions without architectural

modification or task-specific heuristics, enabling MLLMs to acquire robust numerical perception even under severe data imbalance. In summary, our contributions are fourfold.

- **We formulate deep imbalanced regression in MLLMs as a distribution-aware reinforcement learning problem.** We present the first systematic study of DIR under the MLLM paradigm, demonstrating that point-wise numerical supervision—whether via SFT or per-sample regression rewards—fails to capture the global structure of long-tailed continuous targets. Our formulation emphasizes batch-level relational supervision as the key to mitigating regression-to-the-mean behavior.

- **We establish a unified deep imbalanced regression benchmark for MLLMs.** We curate and reformulate four long-tailed numeric prediction datasets into a unified multimodal, dialogue-based benchmark, comprising **over 129k samples in total**, where MLLMs are required to generate continuous values via token-based decoding. We standardize a DIR evaluation protocol by preserving natural long-tailed training distributions and adopting shot-aware balanced test splits, enabling systematic analysis of imbalance effects and fair comparison across MLLM-based methods.

- **We propose a correlation-guided, batch-level reward design for deep imbalanced regression in MLLMs.** We instantiate this design with a Concordance Correlation Coefficient–based reward, which explicitly aligns predicted and ground-truth distributions in terms of correlation, scale, and mean through batch-level relational comparisons. This approach effectively mitigates regression-to-the-mean collapse and improves robustness in sparse and tail regions.

- **Empirical analysis of regression supervision in MLLMs.** Through extensive experiments and ablations, we demonstrate that batch-level, distribution-aware supervision substantially improves stability and accuracy in under-represented regions, establishing a stronger empirical foundation for regression-oriented alignment of MLLMs. Code is publicly available at https://github.com/xmed-lab/CCC-GRPO.

## 2. Related Work

**Deep Imbalanced Regression.** DIR addresses regression problems with highly skewed continuous target distributions. Yang *et al.* (Yang et al., 2021) formally defines this setting and proposes label- and feature-level distribution smoothing to calibrate learning across the target space. Subsequent methods exploit structural consistency between label space and representation space, including ranking-based regularization (Gong et al., 2022), probabilistic modeling with uncertainty (Wang & Wang, 2023), contrastive alignment (Keramati et al.), and group-aware or ordinal formulations (Pu et al., 2025; Xiong & Yao, 2024; Nie et al.). Despite their effectiveness, these methods are developed for *non-generative*, feature-based regression models equipped with explicit continuous prediction heads. They assume direct optimization over real-valued outputs and do not account for the discrete-token generation paradigm underlying MLLMs. As a result, existing DIR methods do not address how tokenized supervision, autoregressive decoding, and sequence-level optimization interact with long-tailed continuous targets in MLLMs.

**Numerical Regression in MLLMs.** Recent evaluations reveal systematic deficiencies of MLLMs in numerical perception, even with model scaling or chain-of-thought prompting (Weng et al., 2025; Chen et al., 2026), highlighting a fundamental mismatch between token-level training objectives and continuous targets. Existing attempts to bridge the discrete–continuous gap in MLLM regression can be broadly categorized into three paradigms. *Architectural modification* methods introduce task-specific tokens or regression heads to enhance numerical precision. Rex-Omni (Jiang et al., 2025) augments the vocabulary with quantized coordinate tokens for object detection, while GEODE (Guo et al., 2025b) activates a dedicated regression head via specialized control tokens for spatial understanding. Although effective, such methods break the unified generative framework of MLLMs and require costly re-alignment to learn the semantics of newly introduced components. *Reasoning-based* approaches reformulate regression as iterative refinement via chain-of-thought reasoning (Wang et al., 2025a; Wu et al., 2025). While expressive, such methods incur substantial inference latency and are ill-suited for perceptual regression tasks. *Loss-level modification* approaches, such as SoftLabel (Wang et al., 2025b), smooth one-hot supervision to encode local numerical proximity. However, these methods remain constrained to token- or digit-level supervision and fail to capture the global magnitude of continuous targets and the effects of long-tailed target distributions. Overall, prior MLLM regression work primarily focuses on improving local/per-sample numerical accuracy for specific domains. The problem of *deep imbalanced regression*—preserving global distributional structure and robustness to rare targets under token-based generation—has not been systematically studied or analyzed.

**Reinforcement Learning for Post-training.** RL has recently emerged as an effective post-training paradigm for LLMs and MLLMs, enabling optimization over sequence-level objectives beyond next-token prediction. Early approaches rely on point-wise scalar rewards, while more recent work has explored relative or group-based supervision to improve robustness and training stability in LLMs. Repre-

sentative works include DISCO, which introduces domain- and difficulty-aware reward scaling to mitigate frequency bias in LLM training (Zhou et al., 2025), and DRO–REBEL, which studies distributionally robust relative-reward learning under preference distribution shifts in LLMs (Sahu & Wells, 2025). GPRS (Zhu et al., 2025) replaces absolute reward magnitudes with group-wise preference comparisons among multiple responses to align optimization with human preference feedback. Other advances focus on improving alignment, stability, or reasoning capability through group-based optimization and progressive RL strategies (Liu et al., 2025a; Zheng et al., 2025; Wu, 2025). Despite their success, these methods are primarily developed and evaluated in *text-only or preference-alignment settings*. They do not explicitly address continuous-valued regression in MLLMs, where models must generate numerical predictions from joint visual–textual inputs under long-tailed target distributions. In particular, existing group-based or relative rewards are not designed to preserve the numerical structure of predictions, which is critical for imbalanced regression. Building on the R1-style paradigm, recent work applies RL to MLLMs for visual reasoning and perception tasks (e.g., Visual-RFT (Liu et al., 2025b), VLM-R1 (Shen et al., 2025), Perception-R1 (Yu et al., 2025)). However, their rewards are typically defined per sample using task-specific discriminative signals, and do not model cross-sample relationships required for deep imbalanced regression, where preserving global numerical structure under skewed targets is essential. Our work is therefore orthogonal to prior RL advances in LLMs/MLLMs. Rather than proposing a new RL algorithm or optimization strategy, we focus on the design of a *batch-level, distribution-aware supervision* that complements existing RL optimizers and provides supervision semantics for long-tailed numerical regression in MLLMs.

## 3. Method

**Preliminaries.** We adopt GRPO (Shao et al., 2024) as the post-training RL framework for MLLMs. GRPO samples multiple generations for the same input and performs policy updates using relative rewards normalized within each generation group, without requiring a learned value critic.

**Task Definition.** We study *DIR* in MLLMs. Given an input instance $(x, c)$ consisting of an image $x$ and a textual prompt $c$, an MLLM with policy $\pi_\theta(\cdot \mid c, x)$ is required to generate a continuous-valued prediction $y \in \mathbb{R}$. In current MLLMs, numerical values are generated autoregressively as discrete token sequences, leading to a fundamental mismatch between *token-level optimization objectives* and the *continuous, ordered nature of regression targets*. As a result, predictions are often optimized independently and biased toward high-density regions under long-tailed supervision. We therefore reformulate numeric prediction in MLLMs as

a *distribution-aware reinforcement learning problem*. Our key insight is that preserving global distributional structure is essential for robust regression under imbalance, rather than optimizing predictions independently.

### 3.1. Distribution-Aware Reinforcement Learning

Unlike SFT, RL enables supervision to be applied directly on decoded numerical values, providing value-level feedback that is invariant to tokenization. Moreover, RL allows flexible reward designs that extend beyond per-sample accuracy. In this work, we exploit this flexibility to introduce *batch-level, distribution-aware rewards* that evaluate each prediction relative to other samples in the same minibatch.

**Multi-Generation Regression Outputs.** For a minibatch of inputs $\{x_1, x_2, \ldots, x_B\}$, GRPO samples $K$ independent generation trajectories for each input. This yields a set of numeric predictions:

$$\mathbf{q}(x_i) = \left[q_1(x_i), q_2(x_i), \ldots, q_K(x_i)\right]^\top, \quad (1)$$

which naturally encode prediction variability. We summarize these outputs using their empirical mean:

$$\mu(x_i) = \frac{1}{K} \sum_{k=1}^{K} q_k(x_i), \quad (2)$$

which provides a stable, low-variance estimate for each compared sample during reward computation.

**Batch-Level Relational Comparison.** Rather than evaluating each prediction against its ground truth in isolation, we construct a relational comparison set within each minibatch, allowing each prediction to be assessed in the context of other samples. For each input $x_i$, the policy generates $K$ stochastic regression outputs $\{q_k(x_i)\}_{k=1}^{K}$. To evaluate the $k$-th sampled prediction $q_k(x_i)$, we construct a batch-level comparison vector by pairing it with the mean predictions of other samples in the minibatch:

$$\mathbf{q}_{i,k} = \left[q_k(x_i), \{\mu(x_j)\}_{j \neq i}\right], \qquad \mathbf{y}_i = \left[y_i, \{y_j\}_{j \neq i}\right], \quad (3)$$

where $j$ indexes samples in the same minibatch with $j \neq i$ and $\mu(x_j)$ represents the empirical mean of the sampled predictions for $x_j$. We use the mean prediction $\mu(x_j)$ as a stable contextual anchor to reduce reward noise and avoid entangling stochastic generations across different samples. Here $y_i$ denotes the scalar ground-truth target of sample $i$, while $\mathbf{y}_i$ denotes the corresponding ground-truth vector used as the reference for batch-level alignment. The elements in $\mathbf{q}_{i,k}$ and $\mathbf{y}_i$ are ordered by the fixed minibatch index to ensure deterministic and reproducible comparison. This construction allows each prediction to be evaluated relative to the empirical distribution of targets observed within the current minibatch, rather than against a single absolute scalar target alone.

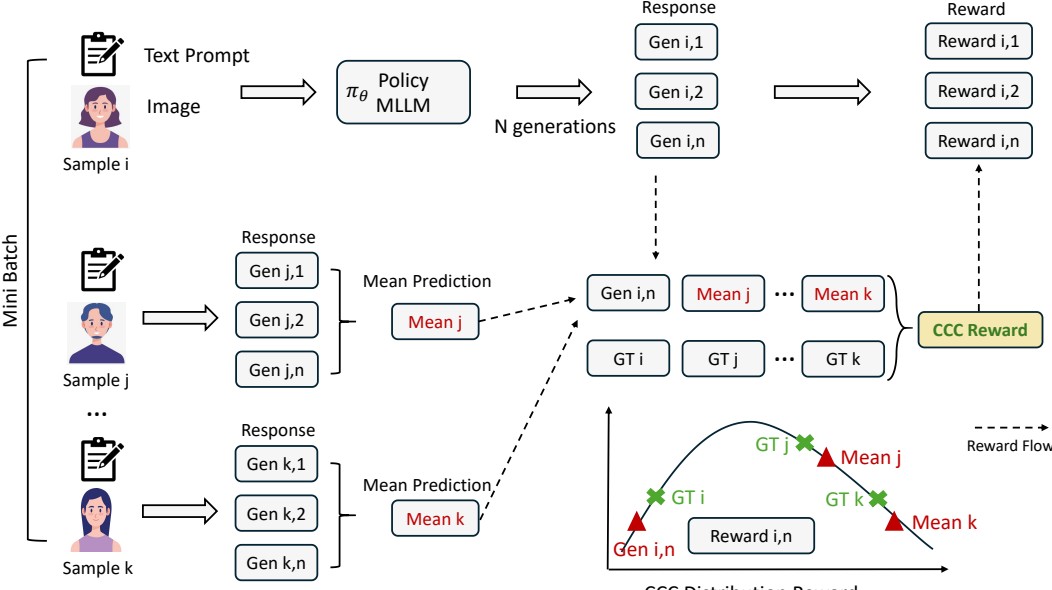

*Figure 3.* Overview of the proposed CCC-GRPO framework for deep imbalanced regression in MLLMs.

**Concordance-Based Distributional Reward.** To quantify the agreement between predicted values and ground-truth targets at the distributional level, we adopt the *concordance correlation coefficient (CCC)* as the reward:

$$\mathrm{CCC}(\mathbf{q}, \mathbf{y}) = \frac{2 \, \mathrm{Cov}(\mathbf{q}, \mathbf{y})}{\mathrm{Var}(\mathbf{q}) + \mathrm{Var}(\mathbf{y}) + \left(\mu_{\mathbf{q}} - \mu_{\mathbf{y}}\right)^2}. \quad (4)$$

CCC simultaneously captures linear correlation, scale consistency, and mean alignment between two distributions (Lawrence & Lin, 1989). Unlike pure correlation or ranking-based objectives, CCC explicitly penalizes both variance collapse and mean shift, making it sensitive to distributional mismatch beyond relative ordering. This property is particularly critical under imbalanced regression settings, where rare target values are otherwise under-emphasized by pointwise loss functions. We define the reward for the $k$-th sampled trajectory of $x_i$ as

$$r_k(x_i) = \mathrm{CCC}(\mathbf{q}_{i,k}, \mathbf{y}_i), \quad (5)$$

We additionally apply a lightweight format validity check reward to ensure stable reward computation. Following standard GRPO, rewards from the $K$ sampled trajectories of each input are normalized within the group to compute relative advantages, which stabilizes policy optimization.

**Summary.** We present a distribution-aware reinforcement learning framework for deep imbalanced regression in MLLMs. By combining GRPO with batch-level CCC rewards, our method provides stable and effective supervision under skewed target distributions, enabling robust numeric prediction without architectural modification.

## 4. Experiments

### 4.1. DIR Benchmark for MLLM

We benchmark CCC-GRPO on a unified suite of deep imbalanced regression tasks designed for MLLMs. All datasets exhibit long-tailed continuous target distributions and are evaluated under a shot-aware protocol. Following standard DIR practice (Yang et al., 2021), training data preserve their naturally imbalanced target distributions, while test sets are constructed to be approximately balanced over the target range. This evaluation setting enables fair and interpretable comparison across dense (many-shot) and sparse (medium/few-shot) regions, and prevents aggregate metrics from being dominated by head-region performance. The benchmark reflects realistic numeric prediction scenarios in which MLLMs are required to directly generate continuous values under severe target imbalance, and supports systematic analysis of imbalance effects as well as fair comparison across MLLM-based methods.

The benchmark consists of four representative regression tasks. *AgeDB-DIR* focuses on age estimation from in-the-wild face images; *IMDB-WIKI-DIR* studies large-scale age prediction from unconstrained web images; *IMDB-Movie-DIR* predicts continuous IMDb ratings from single movie posters, introducing substantial domain shift and label noise; and *BoneAge-DIR* represents a medical quantitative regression task that estimates skeletal maturity from pediatric hand radiographs with inherent label uncertainty. We reconstruct all datasets into a unified DIR benchmark tailored for MLLMs, where models are required to generate continuous values via token-based decoding under naturally skewed training distributions. In total, the benchmark covers over

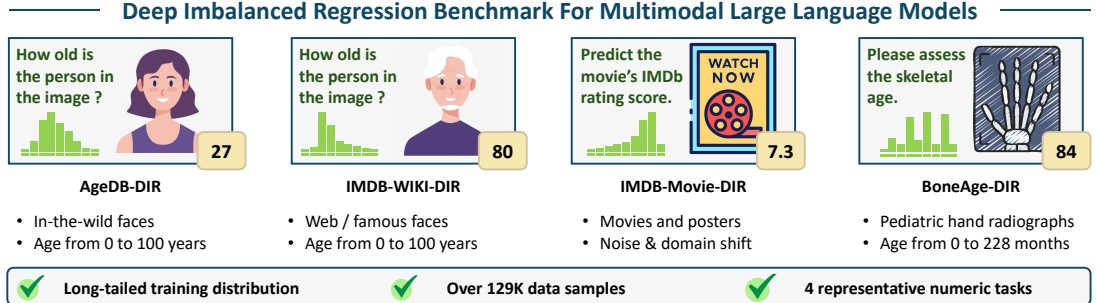

*Figure 4.* Overview of the constructed DIR benchmark for MLLMs.

**129K samples**. Detailed dataset statistics, preprocessing, and split protocols are provided in Appendix B.

**Shot-Aware Evaluation and Metrics.** Following standard DIR practice (Yang et al., 2021), we partition the target space into *many-shot* (over 100 training samples), *medium-shot* (20–100), and *few-shot* (under 20) regions based on training data density. This protocol explicitly evaluates robustness under long-tailed target distributions. We report Mean Absolute Error (MAE) and the Geometric Mean of Absolute Errors (GM). MAE reflects average regression accuracy, while GM penalizes concentrated or frequent errors and provides a complementary measure of error uniformity across sparse and under-represented regions.

**Baselines.** We compare against both classical CNN-based DIR methods and MLLM-based regression approaches. Classical DIR baselines employ continuous regression heads and are reported for reference. Our primary comparisons focus on MLLM-based methods operating under identical backbones, prompting formats, and decoding protocols. All MLLM models are built on Qwen2.5-VL-3B/7B in the main experiments. Complete baseline descriptions and experimental details are provided in Appendix C.

### 4.2. Main Results across Benchmarks

Tables 1–4 summarize results on four representative DIR tasks. Across all datasets, CCC-GRPO (Ours) consistently improves performance in under-represented regions while maintaining competitive accuracy in dense regions, demonstrating the effectiveness of batch-level, distribution-aware supervision for MLLM imbalanced regression. We also provide extended evaluations with sorted error distribution analyses and additional error metrics in Appendix A.

**AgeDB-DIR.** In Table 1, CCC-GRPO achieves the best overall performance among all MLLM-based methods and yields impressive gains in the medium- and few-shot regimes. Compared to SFT, MAE is reduced from 7.67 to 5.62 in the medium-shot region and from 8.36 to 6.40 in the few-shot region under Qwen2.5-VL-3B, while performance in the many-shot region remains stable. This indicates that CCC-

*Table 1.* Benchmarking results on AgeDB-DIR.

| Metrics | MAE ↓ | | | | GM ↓ | | | |
|---|---|---|---|---|---|---|---|---|
| Shot | All | Many | Med. | Few | All | Many | Med. | Few |
| **Deep Imbalanced Regression** | | | | | | | | |
| VANILLA | 7.77 | 6.62 | 9.55 | 13.67 | 5.05 | 4.23 | 7.01 | 10.75 |
| DIR (YANG ET AL., 2021) | 7.47 | 6.69 | 8.30 | 12.55 | 4.71 | 4.25 | 5.36 | 8.59 |
| RANKSIM (GONG ET AL., 2022) | 6.91 | 6.34 | 7.79 | 9.89 | 4.28 | 3.92 | 4.88 | 6.89 |
| VIR (WANG & WANG, 2023) | 6.99 | 6.39 | 7.47 | 9.51 | 4.41 | 4.07 | 5.05 | 6.23 |
| CONR (KERAMATI ET AL.) | 6.81 | 6.32 | 7.45 | 9.21 | 4.39 | 3.81 | 5.01 | 6.02 |
| GROUP-DIR (PU ET AL., 2025) | 6.87 | 6.54 | 6.96 | 9.83 | 4.30 | 4.10 | 4.39 | 6.45 |
| HCA (XIONG & YAO, 2024) | 6.94 | 6.67 | 7.07 | 9.10 | - | - | - | - |
| DIST (NIE ET AL.) | 7.64 | 7.57 | 7.32 | 9.12 | 4.76 | 4.75 | 4.56 | 5.45 |
| **MLLM Regression—Qwen2.5-VL-3B** | | | | | | | | |
| ZEROSHOT | 13.40 | 15.56 | 8.24 | 7.22 | 5.24 | 7.16 | 2.18 | 3.08 |
| SFT | 6.37 | 5.78 | 7.67 | 8.36 | 1.86 | 1.48 | 3.19 | 3.75 |
| SFT-SOFT (WANG ET AL., 2025B) | 6.38 | 5.80 | 7.67 | 8.36 | 1.94 | 1.57 | 3.10 | 3.75 |
| VISUALQUALITY (WU ET AL., 2025) | 9.35 | 9.31 | 10.42 | 6.83 | 3.71 | 3.50 | 4.95 | 2.92 |
| REGRESSION REWARD (TAN ET AL., 2025) | 5.85 | 5.48 | 6.52 | 7.58 | 1.88 | 1.52 | 3.01 | 3.70 |
| DISCO MAE REWARD (ZHOU ET AL., 2025) | 5.95 | 5.64 | 6.73 | 6.75 | 1.84 | 1.59 | 2.53 | 3.00 |
| OURS | 5.52 | 5.42 | 5.62 | 6.40 | 1.63 | 1.43 | 2.07 | 3.04 |
| **MLLM Regression—Qwen2.5-VL-7B** | | | | | | | | |
| ZEROSHOT | 7.16 | 7.05 | 7.17 | 8.23 | 2.61 | 2.42 | 3.05 | 3.42 |
| SFT | 5.82 | 5.35 | 7.03 | 6.98 | 1.59 | 1.18 | 3.57 | 2.73 |
| SFT-SOFT (WANG ET AL., 2025B) | 5.82 | 5.35 | 7.03 | 6.98 | 1.53 | 1.12 | 3.66 | 2.72 |
| VISUALQUALITY (WU ET AL., 2025) | 8.11 | 7.95 | 9.05 | 7.02 | 3.12 | 3.06 | 3.86 | 2.17 |
| REGRESSION REWARD (TAN ET AL., 2025) | 5.62 | 5.41 | 6.12 | 6.22 | 1.67 | 1.45 | 2.25 | 2.92 |
| DISCO MAE REWARD (ZHOU ET AL., 2025) | 5.64 | 5.44 | 6.13 | 6.16 | 1.66 | 1.45 | 2.22 | 2.66 |
| OURS | 5.33 | 5.27 | 5.42 | 5.69 | 1.50 | 1.31 | 2.10 | 2.01 |

GRPO improves generalization beyond dense supervision without degrading head-region accuracy.

**IMDB-Movie-DIR.** In Table 2, CCC-GRPO demonstrates clear advantages in sparse regimes across both model scales, reducing medium shot MAE from 11.21 (SFT) and 10.51 (Regression Reward) to 8.12, and few-shot MAE from 21.51 (SFT) and 21.14 (Regression Reward) to 16.35 under Qwen2.5-VL-3B. Beyond MAE, CCC-GRPO also achieves strong GM performance in the medium- and few-shot regions, indicating more uniform error distributions and improved robustness under noisy and under-represented targets. Unlike point-wise regression rewards, which exhibit unstable behavior across shot regimes, CCC-GRPO delivers improved performance by enforcing distribution-level consistency rather than independent numeric fitting. Figure 5 further shows that the performance gains are concentrated in low-density target ranges, while performance in the many-

*Table 2.* Benchmarking results on IMDB-Movie-DIR.

| Metrics | MAE ↓ | | | | GM ↓ | | | |
|---|---|---|---|---|---|---|---|---|
| Shot | All | Many | Med. | Few | All | Many | Med. | Few |
| **MLLM Regression—Qwen2.5-VL-3B** | | | | | | | | |
| ZeroShot | 13.63 | 14.17 | 10.61 | 18.99 | 7.24 | 9.66 | 2.47 | 16.81 |
| SFT | 7.44 | 4.87 | 11.21 | 21.51 | 2.57 | 1.52 | 7.02 | 18.28 |
| SFT-Soft (Wang et al., 2025b) | 7.43 | 4.87 | 11.15 | 21.57 | 2.52 | 1.52 | 6.60 | 18.33 |
| VisualQuality (Wu et al., 2025) | 24.43 | 24.44 | 25.25 | 20.23 | 14.75 | 13.71 | 18.44 | 13.42 |
| Regression Reward | 7.42 | 5.06 | 10.51 | 21.14 | 3.13 | 1.96 | 7.46 | 18.68 |
| DISCO MAE Reward (Zhou et al., 2025) | 7.53 | 5.36 | 10.38 | 20.61 | 3.02 | 2.05 | 5.91 | 16.89 |
| Ours | 6.89 | 5.60 | 8.12 | 16.35 | 2.11 | 1.63 | 3.24 | 7.17 |
| **MLLM Regression—Qwen2.5-VL-7B** | | | | | | | | |
| ZeroShot | 10.42 | 7.18 | 17.66 | 18.91 | 3.43 | 2.48 | 7.72 | 5.86 |
| SFT | 6.42 | 4.36 | 9.58 | 17.14 | 2.01 | 1.21 | 5.69 | 10.88 |
| SFT-Soft (Wang et al., 2025b) | 6.43 | 4.38 | 9.56 | 17.16 | 1.93 | 1.14 | 5.66 | 11.04 |
| VisualQuality (Wu et al., 2025) | 24.73 | 23.38 | 28.88 | 24.04 | 12.35 | 10.30 | 19.24 | 17.31 |
| Regression Reward (Tan et al., 2025) | 6.23 | 4.36 | 8.64 | 17.73 | 1.69 | 0.95 | 5.12 | 13.90 |
| DISCO MAE Reward (Zhou et al., 2025) | 6.46 | 4.46 | 9.14 | 18.35 | 1.89 | 1.07 | 5.77 | 14.71 |
| Ours | 5.95 | 4.85 | 6.87 | 14.58 | 1.62 | 1.33 | 1.92 | 7.20 |

shot region remains competitive.

**IMDB-WIKI-DIR.** This dataset exhibits extreme long-tailed distributions with multiple sparsely populated target bins. In Table 3, CCC-GRPO achieves the best overall performance among all MLLM-based methods, attaining the lowest MAE in nearly all shot regimes and the strongest GM performance in the many- and medium-shot regions, demonstrating robust behavior under severe imbalance across the full target spectrum.

**BoneAge-DIR.** Table 4 reports the results across Qwen2.5-VL-3B/7B. CCC-GRPO achieves the lowest MAE and produces more uniform error distributions across shot regimes. As illustrated by the sorted error curves in Figure 6, CCC-GRPO significantly suppresses extreme errors compared to SFT, particularly in sparse regions. We observe that GM may increase slightly in certain many-shot settings, which is expected given the multiplicative nature of GM and its sensitivity to frequent small deviations; detailed analysis is provided in Appendix A.

**Task Difficulty.** For natural-image regression tasks (e.g., age and movie poster), SFT already achieves strong overall performance, and CCC-GRPO mainly improves robustness in long-tailed regions across both 3B and 7B models. In contrast, BoneAge-DIR is substantially more challenging, with poor zero-shot performance, where CCC-GRPO yields a much larger gain, achieving a +23.55% overall MAE improvement over SFT shown in Table 13. Notably, despite the *multi-peaked training label distribution* of BoneAge-DIR (Fig 13), CCC-GRPO remains highly effective, highlighting its advantage on harder and more structurally complex regression problems.

### 4.3. Ablation Study

We conduct ablation studies primarily on AgeDB-DIR under Qwen2.5-VL-3B, which provides a controlled setting to isolate the effects of reward design, supervision scope, and

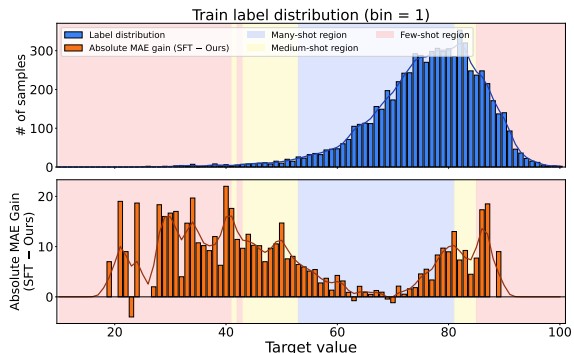

*Figure 5.* MAE gain of Ours over SFT on IMDB-Movie-DIR under Qwen2.5-VL-3B.

*Table 3.* Benchmarking results on IMDB-WIKI-DIR.

| Metrics | MAE ↓ | | | | GM ↓ | | | |
|---|---|---|---|---|---|---|---|---|
| Shot | All | Many | Med. | Few | All | Many | Med. | Few |
| **Deep Imbalanced Regression** | | | | | | | | |
| Vanilla | 8.06 | 7.23 | 15.12 | 26.33 | 4.57 | 4.17 | 10.59 | 20.46 |
| DIR (Yang et al., 2021) | 7.78 | 7.20 | 12.61 | 22.19 | 4.37 | 4.12 | 7.39 | 12.61 |
| RankSim (Gong et al., 2022) | 7.35 | 6.81 | 11.50 | 22.75 | 4.05 | 3.85 | 6.05 | 14.68 |
| VIR (Wang & Wang, 2023) | 7.19 | 6.56 | 11.81 | 20.96 | 3.85 | 3.63 | 6.51 | 12.23 |
| ConR (Keramati et al.) | 7.33 | 6.69 | 11.87 | 21.53 | 3.99 | 3.81 | 6.66 | 12.62 |
| Group-DIR (Pu et al., 2025) | 7.22 | 6.71 | 11.42 | 20.25 | 3.88 | 3.68 | 5.74 | 11.13 |
| HCA (Xiong & Yao, 2024) | 7.31 | 6.82 | 10.99 | 22.04 | - | - | - | - |
| Dist (Nie et al.) | 8.02 | 7.48 | 12.30 | 22.33 | 4.59 | 4.37 | 7.08 | 13.02 |
| **MLLM Regression—Qwen2.5-VL-3B** | | | | | | | | |
| ZeroShot | 9.44 | 9.09 | 11.59 | 24.09 | 3.46 | 3.38 | 3.84 | 10.92 |
| SFT | 7.10 | 6.53 | 11.73 | 21.88 | 1.81 | 1.60 | 5.89 | 11.84 |
| SFT-Soft (Wang et al., 2025b) | 7.10 | 6.53 | 11.72 | 21.88 | 1.84 | 1.63 | 5.88 | 11.88 |
| VisualQuality (Wu et al., 2025) | 16.12 | 15.99 | 17.13 | 20.36 | 9.12 | 8.84 | 13.02 | 9.18 |
| Regression Reward (Tan et al., 2025) | 7.17 | 6.60 | 11.68 | 23.06 | 1.92 | 1.77 | 3.77 | 13.26 |
| DISCO MAE Reward (Zhou et al., 2025) | 7.13 | 6.66 | 10.88 | 20.78 | 1.83 | 1.68 | 3.91 | 9.94 |
| Ours | 6.71 | 6.26 | 9.89 | 20.17 | 1.60 | 1.53 | 2.06 | 12.35 |
| **MLLM Regression—Qwen2.5-VL-7B** | | | | | | | | |
| ZeroShot | 7.42 | 6.95 | 10.90 | 23.06 | 2.11 | 1.98 | 3.38 | 16.53 |
| SFT | 6.61 | 6.11 | 10.56 | 20.44 | 1.46 | 1.32 | 3.65 | 9.86 |
| SFT-Soft (Wang et al., 2025b) | 6.61 | 6.11 | 10.52 | 20.44 | 1.44 | 1.30 | 3.47 | 9.91 |
| VisualQuality (Wu et al., 2025) | 8.56 | 8.33 | 10.01 | 18.29 | 2.96 | 2.90 | 3.42 | 5.40 |
| Regression Reward (Tan et al., 2025) | 6.62 | 6.13 | 10.51 | 20.64 | 1.45 | 1.34 | 2.79 | 12.03 |
| DISCO MAE Reward (Zhou et al., 2025) | 6.64 | 6.13 | 10.77 | 20.59 | 1.41 | 1.30 | 2.91 | 13.28 |
| Ours | 6.41 | 5.95 | 9.96 | 19.94 | 1.33 | 1.25 | 2.06 | 10.31 |

optimization choices.

**Effect of Reward Design and Supervision Scope.** Table 5 compares different reward formulations under a unified GRPO framework. Importantly, this ablation reflects not only the choice of reward function, but also the *scope of supervision*: point-wise rewards (e.g., MAE) operate at the per-sample level, while Spearman and CCC introduce batch-level relational comparison. Per-sample regression rewards substantially reduce overall error compared to SFT, but remain biased toward dense regions and exhibit limited robustness in sparse regimes. DISCO MAE Reward corresponds to our reproduction of difficulty-aware reweighting (Zhou et al., 2025), adapted to the generative numeric regression setting of MLLMs. It further improves tail performance by adjusting instance importance, yet still relies on point-wise supervision and does not explicitly model inter-sample structure. In contrast, batch-level relational rewards (Spearman

*Table 4.* Benchmarking results on BoneAge-DIR.

| Metrics | MAE↓ | | | | GM↓ | | | |
|---|---|---|---|---|---|---|---|---|
| Shot | All | Many | Med. | Few | All | Many | Med. | Few |
| **MLLM Regression—Qwen2.5-VL-3B** | | | | | | | | |
| ZeroShot | 95.36 | 112.61 | 82.90 | 79.98 | 61.44 | 104.10 | 59.52 | 19.14 |
| SFT | 18.60 | 17.53 | 18.22 | 21.86 | 5.16 | 2.56 | 7.85 | 11.39 |
| SFT-Soft (Wang et al., 2025b) | 18.70 | 17.59 | 18.39 | 21.91 | 5.23 | 2.63 | 7.94 | 11.38 |
| VisualQuality (Wu et al., 2025) | 39.13 | 39.30 | 42.06 | 32.86 | 20.57 | 12.86 | 32.69 | 24.38 |
| Regression Reward (Tan et al., 2025) | 15.45 | 14.87 | 15.14 | 17.42 | 3.86 | 2.13 | 4.79 | 9.99 |
| DISCO MAE Reward (Zhou et al., 2025) | 20.87 | 21.07 | 19.31 | 23.51 | 6.21 | 5.54 | 5.29 | 11.16 |
| Ours | 14.22 | 14.15 | 14.22 | 14.35 | 4.74 | 4.17 | 4.30 | 7.80 |
| **MLLM Regression—Qwen2.5-VL-7B** | | | | | | | | |
| ZeroShot | 93.69 | 110.65 | 81.34 | 78.76 | 59.70 | 101.95 | 57.17 | 18.68 |
| SFT | 17.09 | 16.20 | 16.82 | 19.71 | 3.87 | 1.67 | 5.40 | 14.30 |
| SFT-Soft (Wang et al., 2025b) | 17.01 | 16.10 | 16.61 | 19.90 | 3.82 | 1.62 | 5.34 | 14.44 |
| VisualQuality (Wu et al., 2025) | 37.78 | 36.94 | 41.14 | 33.04 | 19.38 | 11.53 | 30.99 | 25.50 |
| Regression Reward (Tan et al., 2025) | 15.00 | 14.47 | 14.79 | 16.66 | 3.43 | 1.96 | 4.00 | 9.35 |
| DISCO MAE Reward (Zhou et al., 2025) | 20.17 | 18.66 | 20.34 | 23.36 | 5.93 | 2.56 | 9.19 | 17.44 |
| Ours | 13.35 | 13.35 | 13.69 | 12.67 | 3.08 | 2.30 | 3.31 | 5.26 |

*Figure 6.* Sorted error distribution curves for CCC-GRPO and SFT on the BoneAge-DIR dataset under Qwen2.5-VL-3B.

and CCC) consistently improve performance in medium- and few-shot regions, confirming the importance of cross-sample comparison under long-tailed distributions. Among them, CCC achieves the most balanced performance across all shot regimes. While Spearman rewards effectively preserve relative ordering and improve few-shot accuracy, they lack explicit constraints on absolute scale and mean alignment, leading to suboptimal performance in dense regions. By jointly enforcing correlation, scale, and mean consistency, CCC provides a more complete distribution-aware supervision signal, resulting in the best overall trade-off. We provide a detailed analysis in Appendix D.

*Table 5.* Ablation Study of Reward Design.

| Metrics | MAE↓ | | | | GM↓ | | | |
|---|---|---|---|---|---|---|---|---|
| Shot | All | Many | Med. | Few | All | Many | Med. | Few |
| PAIRWISE RANK REWARD | 9.35 | 9.31 | 10.42 | 6.83 | 3.71 | 3.50 | 4.95 | 2.92 |
| +OURS | 5.57 | 5.51 | 5.61 | 6.01 | 1.68 | 1.56 | 1.87 | 2.50 |
| REGRESSION MAE REWARD | 5.85 | 5.48 | 6.52 | 7.58 | 1.88 | 1.52 | 3.01 | 3.70 |
| +OURS | 5.61 | 5.63 | 5.37 | 6.12 | 1.68 | 1.58 | 1.63 | 3.15 |
| SPEARMAN CORRELATION REWARD | 6.81 | 7.26 | 5.81 | 5.37 | 2.52 | 2.68 | 1.83 | 3.40 |
| +OURS | 5.68 | 5.45 | 5.62 | 6.52 | 1.69 | 1.51 | 1.82 | 4.18 |
| DISCO MAE REWARD | 5.95 | 5.64 | 6.73 | 6.75 | 1.84 | 1.59 | 2.53 | 3.00 |
| OURS | 5.52 | 5.42 | 5.62 | 6.40 | 1.63 | 1.43 | 2.07 | 3.04 |

**Robustness across GRPO Variants.** Table 6 evaluates CCC-based rewards under different GRPO variants, including DrGRPO (Liu et al., 2025a) and RegGRPO (Park et al., 2025). Performance differences are minor, indicating that the observed gains are largely insensitive to the specific policy optimization strategy. This confirms our performance improvements primarily stem from the *reward design* rather than from modifications to the RL algorithm itself.

*Table 6.* Ablation Study of RL Variants with and without CCC.

| Metrics | MAE↓ | | | | GM↓ | | | |
|---|---|---|---|---|---|---|---|---|
| Shot | All | Many | Med. | Few | All | Many | Med. | Few |
| DrGRPO (Liu et al., 2025a) | 6.08 | 5.65 | 6.99 | 7.72 | 1.97 | 1.56 | 3.52 | 3.51 |
| DrGRPO+CCC | 5.73 | 5.59 | 5.97 | 6.36 | 1.58 | 1.35 | 2.05 | 3.33 |
| RegGRPO (Park et al., 2025) | 6.00 | 5.53 | 7.10 | 7.44 | 1.91 | 1.61 | 2.61 | 3.87 |
| RegGRPO+CCC | 5.56 | 5.41 | 5.81 | 6.25 | 1.59 | 1.51 | 1.60 | 2.65 |
| GRPO (Shao et al., 2024) | 5.85 | 5.48 | 6.52 | 7.58 | 1.88 | 1.52 | 3.01 | 3.70 |
| GRPO+CCC | 5.52 | 5.42 | 5.62 | 6.40 | 1.63 | 1.43 | 2.07 | 3.04 |

**Sensitivity to Number of Generations and Batch Size.** In Table 7, CCC-GRPO exhibits stable performance across different numbers of sampled generations and batch sizes, with no sharp degradation observed when varying either factor. In particular, using a small number of generations already yields competitive performance, suggesting CCC-GRPO can operate effectively without requiring extensive sampling. Similarly, performance remains relatively robust under different batch sizes, indicating the proposed batch-level reward is not overly sensitive to batch configuration.

*Table 7.* Ablation Study of Number of Generations / Batch Size.

| Metrics | MAE↓ | | | | GM↓ | | | |
|---|---|---|---|---|---|---|---|---|
| Shot | All | Many | Med. | Few | All | Many | Med. | Few |
| 4 GENERATION | 5.52 | 5.42 | 5.62 | 6.40 | 1.63 | 1.43 | 2.07 | 3.04 |
| 6 GENERATION | 5.53 | 5.32 | 5.94 | 6.31 | 1.67 | 1.43 | 2.15 | 3.55 |
| 8 GENERATION | 5.53 | 5.39 | 5.73 | 6.34 | 1.45 | 1.28 | 1.56 | 3.92 |
| 16 BATCH SIZE | 5.52 | 5.42 | 5.62 | 6.40 | 1.63 | 1.43 | 2.07 | 3.04 |
| 24 BATCH SIZE | 5.52 | 5.44 | 5.48 | 6.37 | 1.53 | 1.38 | 1.50 | 4.43 |
| 28 BATCH SIZE | 5.55 | 5.44 | 5.73 | 6.10 | 1.69 | 1.54 | 1.82 | 3.37 |
| 32 BATCH SIZE | 5.56 | 5.54 | 5.49 | 5.98 | 1.52 | 1.40 | 1.56 | 3.08 |

**Robustness Across Random Seeds.** Table 8 reports mean±std results over five random seeds for different batch sizes on AgeDB-DIR. The variance across runs remains small, and the overall performance changes only marginally across batch-size settings. These results confirm that the proposed method is stable across random seeds and robust to stochastic mini-batch composition.

## 5. Conclusions

We study deep imbalanced regression in multimodal large language models and reveal a fundamental limitation of point-wise, token-level supervision when continuous targets follow long-tailed distributions. Such objectives bias optimization toward dense regions, encourage regression-to-the-mean behavior, and fail to preserve global numeric structure, leading to unreliable predictions in under-represented regimes. To address this challenge, we propose a reinforcement learning framework that shifts supervision from isolated, per-sample errors to *batch-level relational comparison*. Built on GRPO, our approach enables MLLMs to learn distributional structure directly from cross-sample relationships, without architectural modification or task-specific heuristics. Instantiated with a CCC-based reward, the pro-

Table 8. Multi-seed batch-size ablation on AgeDB-DIR. We report mean±std over five random seeds.

| | MAE↓ | | | |
|---|---|---|---|---|
| Batch | All | Many | Med. | Few |
| 24 | 5.56±0.02 | 5.46±0.04 | 5.68±0.15 | 6.13±0.16 |
| 28 | 5.58±0.03 | 5.44±0.07 | 5.77±0.10 | 6.30±0.19 |
| 32 | 5.58±0.02 | 5.57±0.04 | 5.46±0.16 | 5.97±0.18 |
| | GM↓ | | | |
| Batch | All | Many | Med. | Few |
| 24 | 1.50±0.08 | 1.35±0.10 | 1.71±0.19 | 2.90±0.86 |
| 28 | 1.58±0.08 | 1.44±0.10 | 1.73±0.18 | 3.13±0.42 |
| 32 | 1.63±0.08 | 1.56±0.11 | 1.58±0.33 | 3.03±0.40 |

posed method jointly aligns correlation, scale, and mean between predictions and targets, effectively mitigating prediction collapse and substantially improving robustness in medium- and few-shot regions. Extensive experiments and ablation studies demonstrate that the primary driver of performance gains is the use of *distribution-aware, batch-level supervision itself*. This observation suggests that batch-level comparison constitutes a powerful and general supervision paradigm for long-tailed regression in generative models. While our experiments cover only a limited range of numeric regression scenarios, the proposed framework is broadly applicable to a wide range of numeric prediction problems in MLLMs. We hope this work encourages further exploration of distribution-aware reinforcement learning for reliable numeric prediction, particularly in safety-critical and severely imbalanced settings.

## Acknowledgments

This work was supported by a research grant from the Joint Research Scheme (JRS) under the National Natural Science Foundation of China (NSFC) and the Research Grants Council (RGC) of Hong Kong (Project No. N_HKUST654/24); a research grant from the RGC of the Hong Kong Special Administrative Region, China (Project No. R6005-24); and a research grant from the Innovation and Technology Commission (ITC) (Project No. GHP/124/22).

## Impact Statement

This work studies distribution-aware reinforcement learning for deep imbalanced regression in multimodal large language models. While the proposed method improves performance under long-tailed target distributions, it also raises important considerations regarding fairness and safety. First, datasets such as AgeDB and IMDB-WIKI may exhibit correlations between long-tailed regions and underrepresented demographic groups. Optimizing distribution-level objectives does not guarantee equitable performance across subpopulations and may unintentionally shift error distributions. Therefore, practical deployment should include subgroup-

level evaluation, fairness auditing, and calibration analysis rather than relying solely on aggregate metrics. Second, in safety-critical applications such as medical prediction (e.g., BoneAge estimation), reliability on common cases is essential. Distribution-level objectives may introduce undesirable trade-offs if used without additional constraints. We therefore view our method as a training-time research strategy rather than a deployment prescription. In such settings, real-world use would require additional safeguards, including stronger instance-level constraints, clinically relevant subgroup analysis, and failure-case auditing. Overall, we believe fairness and safety evaluation should be treated as necessary prerequisites for real-world deployment.

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

# Supplementary Material

## A. Extended Experimental Results

**Sorted Error Distribution Analysis.**  Figures 7–10 present sorted absolute error curves for CCC-GRPO and supervised fine-tuning (SFT) across all evaluated datasets under Qwen2.5-VL-3B. Each curve visualizes the per-sample absolute errors on the test set, sorted in ascending order. The rightmost region of each curve therefore corresponds to the worst-performing samples, which mainly correspond to samples from few-shot or under-represented target regions.

Across all datasets, the sorted error curves of CCC-GRPO consistently lie below those of SFT in the medium- and few-shot regions, indicating lower per-sample errors and more stable behavior on under-represented targets. While the two methods achieve comparable accuracy on low-error (many-shot) samples, CCC-GRPO markedly suppresses extreme errors in the high-error regime. These results suggest that the performance gains of CCC-GRPO do not stem from uniform improvements across all samples, but rather from reducing large deviations on difficult or rare cases. Such tail-focused error reduction is not fully reflected by average metrics alone and provides complementary evidence for the effectiveness of batch-level, distribution-aware supervision under long-tailed distributions.

Figure 11–12 further visualize MAE gains with respect to training data density. For all datasets, improvements over SFT are concentrated in low- and medium-density regions, while performance in dense regions remains comparable. This pattern aligns with the intended behavior of CCC-GRPO: mitigating regression collapse in sparse regimes without explicitly sacrificing head-region accuracy.

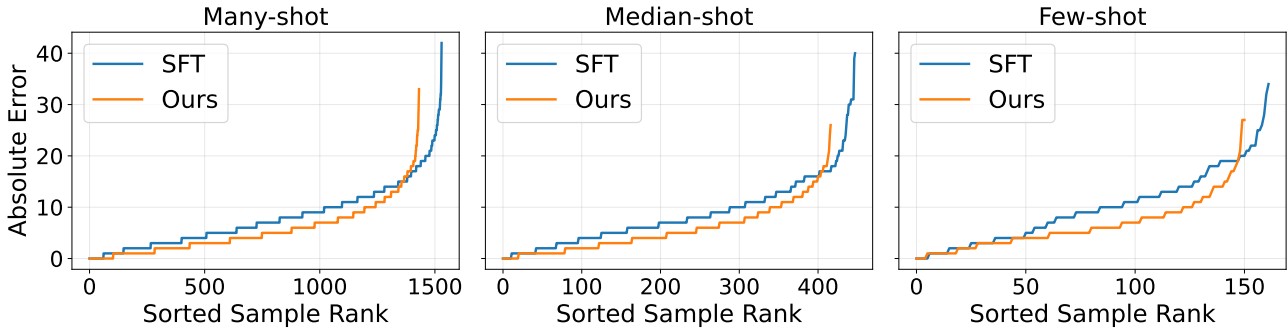

*Figure 7.* Sorted error distribution curves for CCC-GRPO and SFT on the AgeDB-DIR dataset

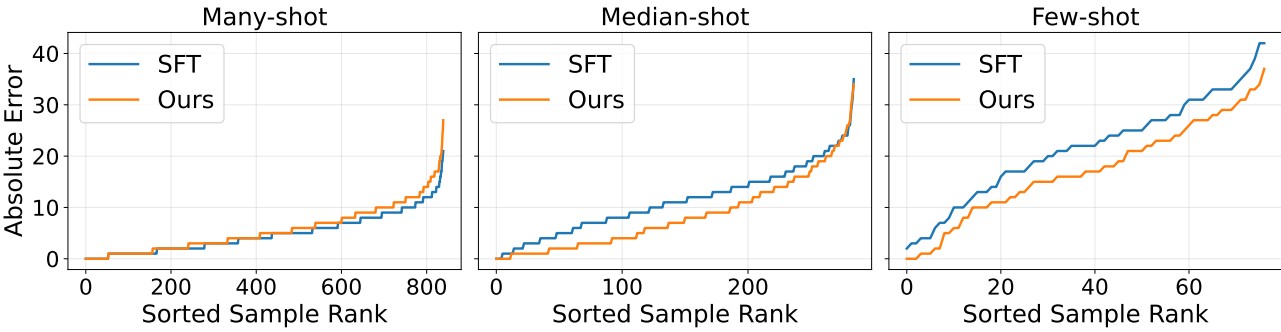

*Figure 8.* Sorted error distribution curves for CCC-GRPO and SFT on the IMDB-Movie-DIR dataset

**Complementary Error Metrics.**  Tables 10–13 report detailed results using three complementary metrics: Mean Squared Error (MSE), Mean Absolute Error (MAE), and the Geometric Mean of Absolute Errors (GM). MSE amplifies large errors and is therefore sensitive to catastrophic failures, MAE reflects average prediction accuracy, while GM penalizes frequent moderate deviations through multiplicative aggregation.

Across datasets, CCC-GRPO achieves consistently strong performance under all three metrics, with clear advantages in medium- and few-shot regions. The amplified nature of MSE further highlights the benefits of distribution-aware supervision:

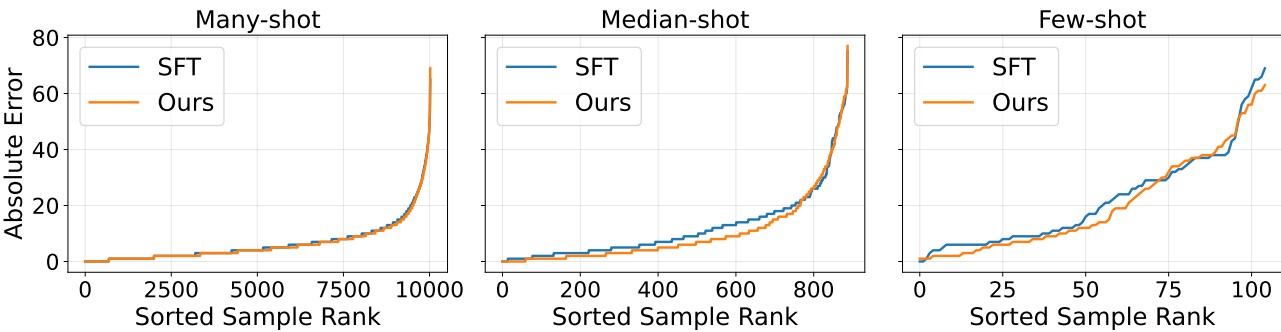

*Figure 9.* Sorted error distribution curves for CCC-GRPO and SFT on the IMDB-WIKI-DIR dataset

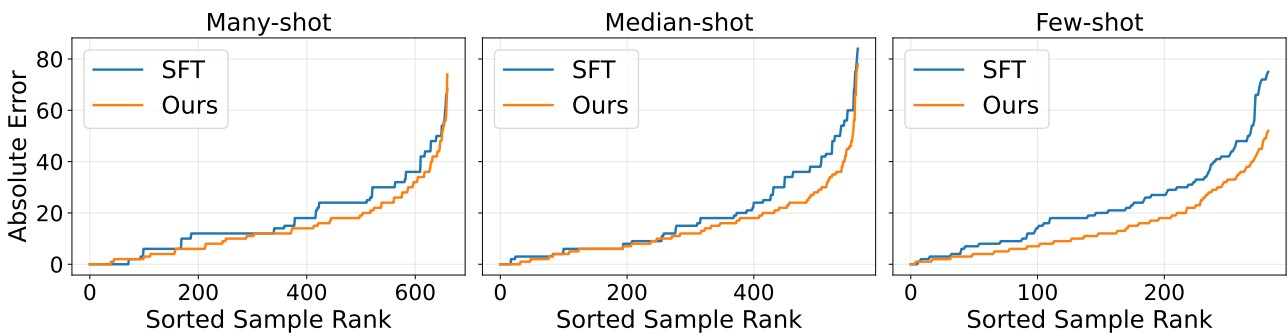

*Figure 10.* Sorted error distribution curves for CCC-GRPO and SFT on the BoneAge-DIR dataset

by reducing extreme deviations, CCC-GRPO exhibits more pronounced gains under MSE than under MAE, especially in sparse regions. These trends are consistent with the sorted error curves, confirming that CCC-GRPO improves regression robustness by stabilizing predictions across the target spectrum rather than optimizing mean accuracy alone.

**Head–Tail Trade-off under Extreme Imbalance.** On IMDB-Movie-DIR, we observe a mild degradation in the many-shot region. This reflects an inherent trade-off of distribution-aware objectives: discouraging prediction collapse and preserving global variance improves tail reliability, but may slightly reduce mean-optimal accuracy in extremely dense regions (Nie et al.; Pu et al., 2025). We view this trade-off as practically meaningful, especially for applications where rare cases carry disproportionate importance (e.g., clinical imaging). Exploring hybrid objectives that explicitly balance head and tail performance is an interesting direction for future work.

**Interpreting GM Degradation on BoneAge-DIR.** On BoneAge-DIR, CCC-GRPO improves MAE and yields more uniform performance across shot regions, but results in higher GM error in many-shot regions in Table 4 and Table 13. This behavior can be attributed to the intrinsic sensitivity of GM, which aggregates errors multiplicatively and can be influenced by moderately larger deviations in a subset of samples. Prior work on deep imbalanced regression has shown that methods with improved overall error distributions may nonetheless exhibit worse GM when errors at a few ranked positions increase slightly (Nie et al.). In BoneAge-DIR, label ambiguity—particularly in adolescent age ranges—further amplifies this effect.

Importantly, the observed GM increase does not imply inferior overall behavior in dense regions. As shown in Figure 10, within the many-shot subset, the sorted error curve of CCC-GRPO lies consistently below that of SFT over the majority of samples, indicating lower absolute errors for most instances. The higher GM value is therefore driven by a small fraction of samples near the extreme tail, rather than a uniform degradation or systematic shift of the error distribution. This phenomenon reflects a metric–objective mismatch rather than a failure of the proposed framework. CCC-GRPO explicitly optimizes distributional alignment (correlation, scale, and mean), whereas GM emphasizes multiplicative aggregation of errors. In high-uncertainty or noisy-label settings, these objectives may not be perfectly aligned, representing an inherent trade-off when preserving global variance and distributional structure.

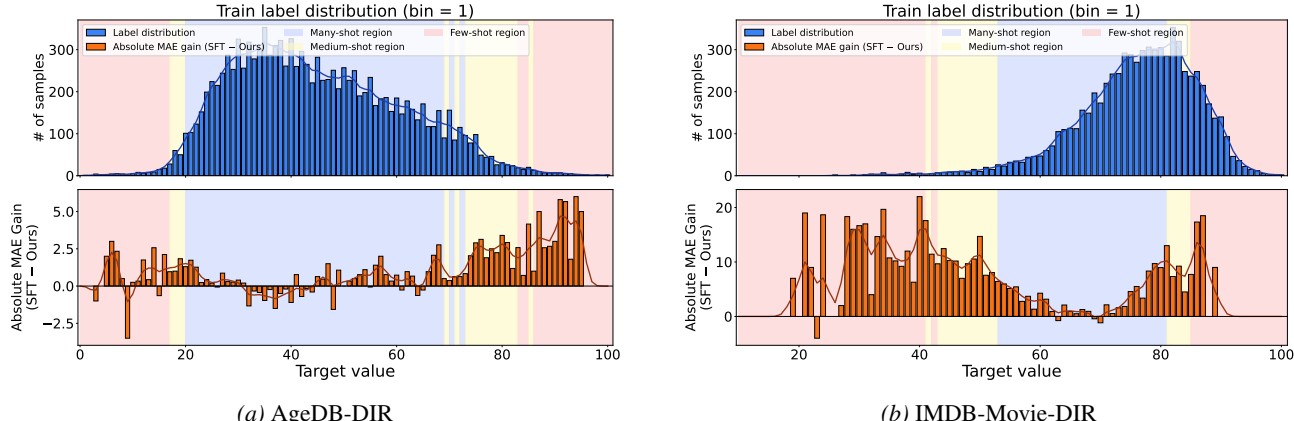

*(a)* AgeDB-DIR

*(b)* IMDB-Movie-DIR

*Figure 11.* MAE gain across AgeDB-DIR and IMDB-Movie-DIR datasets under imbalanced training distributions.

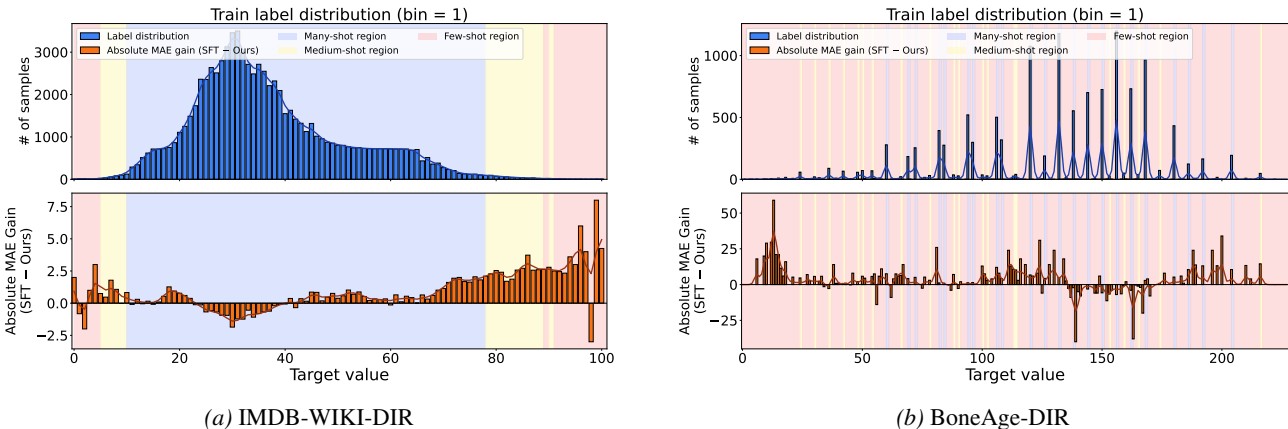

*(a)* IMDB-WIKI-DIR

*(b)* BoneAge-DIR

*Figure 12.* MAE gain across IMDB-WIKI-DIR and BoneAge-DIR datasets under imbalanced training distributions.

**Scaling to Larger MLLM Backbones.** In addition to the main experiments on Qwen2.5-VL-3B, we further evaluate CCC-GRPO on a larger backbone, Qwen2.5-VL-7B, with detailed results reported in Table 10–13. Across all four datasets, the 3B and 7B models exhibit consistent performance trends. After scaling the backbone from 3B to 7B, the relative behavior across methods remains qualitatively similar: CCC-GRPO continues to improve robustness in medium- and few-shot regions, suppress extreme errors, and maintain competitive performance in dense regions. This trend is observed consistently on AgeDB-DIR, IMDB-Movie-DIR, IMDB-WIKI-DIR, and BoneAge-DIR. Due to computational constraints, we do not evaluate substantially larger model sizes in this work. We leave a more systematic investigation of model scaling effects, together with an expanded set of challenging numeric regression benchmarks, as an important direction for future work.

## B. Dataset Construction and Imbalance Characteristics

All benchmarks use naturally imbalanced training distributions. Test sets are constructed to be approximately balanced over the supported target range. Figures 13 and 14 visualize the target distributions of the training and testing sets, respectively, highlighting the long-tailed imbalance in training and the approximately balanced evaluation protocol used at test time.

### B.1. AgeDB-DIR

AgeDB-DIR is constructed from AgeDB (Moschoglou et al., 2017), a manually curated in-the-wild age dataset with accurate labels. We preserve the naturally imbalanced training distribution and construct balanced test set following standard DIR practice (Yang et al., 2021). The training set contains 12,208 images with ages ranging from 0 to 100. Using 1-year bins, the maximum bin density is 353 and the minimum bin density is 1. The test set is balanced across age bins, containing 2,140 images.

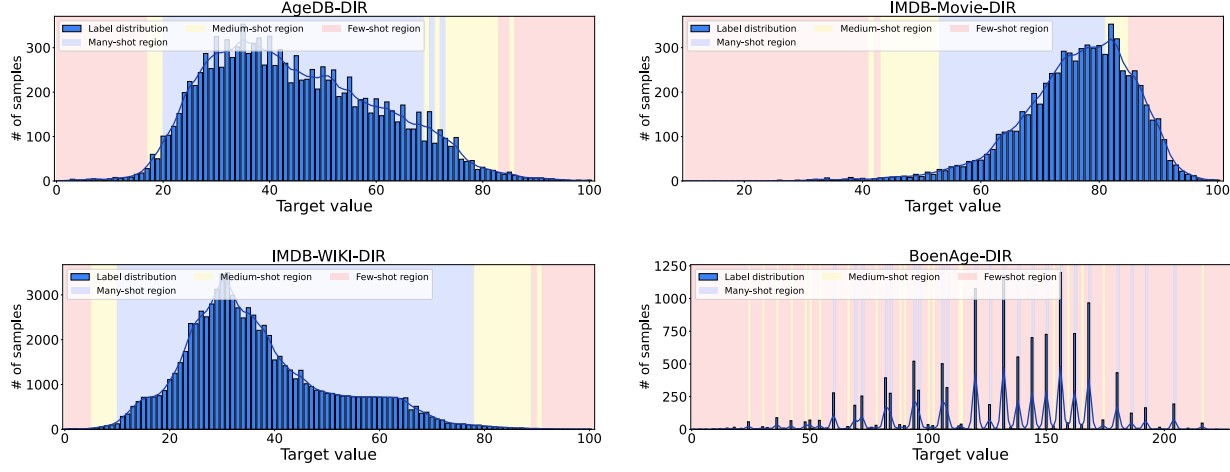

*Figure 13.* Imbalanced Training Dataset Overview

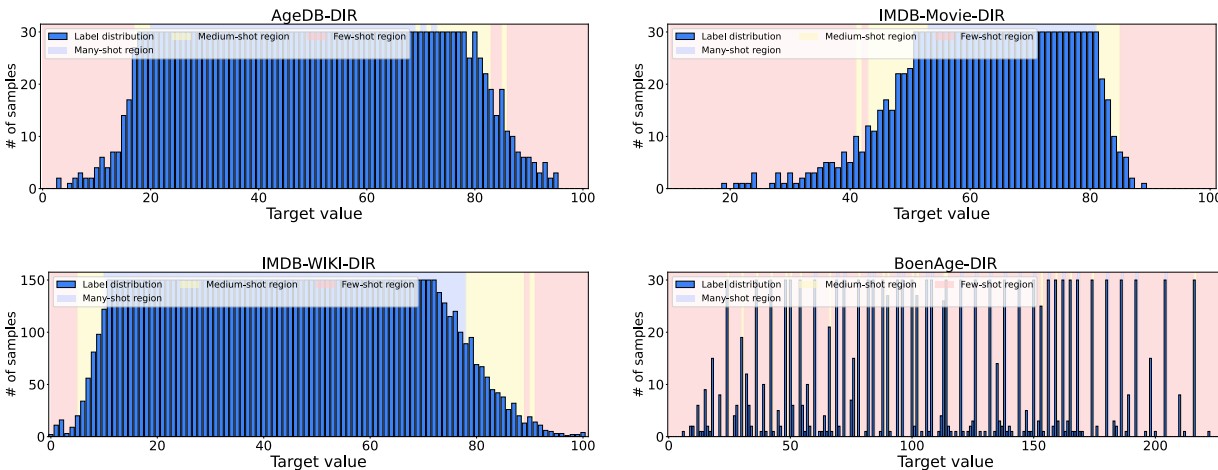

*Figure 14.* Balanced Testing Dataset Overview

### B.2. IMDB-Movie-DIR

IMDB-Movie-DIR is constructed from the IMDB movie dataset (Kaggle, 2025), where each sample consists of a single movie poster paired with a continuous IMDb rating score. The task requires predicting the movie rating from visual input only, introducing substantial domain shift and label noise. We preserve the naturally imbalanced training distribution, which is heavily concentrated in mid-range scores and sparse at both extremes. For numerical stability and clearer comparison, rating values are scaled by a factor of 10 during training and evaluation, without affecting relative performance. The dataset contains 7,049 training samples and 1,203 samples for testing. The test set is approximately balanced across the rating range to enable shot-aware evaluation.

### B.3. IMDB-WIKI-DIR

IMDB-WIKI-DIR is derived from IMDB-WIKI (Rothe et al., 2018). After filtering low-quality images, the curated dataset contains 191.5K training images and 11.0K images for validation and testing, with ages ranging from 0 to 100 and extreme imbalance (1–7,000 samples per age bin) (Yang et al., 2021). During training, we downsample the original training set according to the *original* (imbalanced) distribution shape to improve computational efficiency. After downsampling, the dataset contains 81,911 training samples and 11,016 samples for testing. Despite this, imbalance remains severe: the maximum bin count still exceeds 3,500, the minimum bin count is 1. The test set is constructed to be approximately balanced over the supported age range, enabling fair evaluation across dense and sparse regions.

*Table 9.* Summary of the MLLM Deep Imbalanced Regression (DIR) benchmarks. All datasets use naturally imbalanced training distributions and balanced test splits; no validation sets are used.

| Dataset | Train | Test | Total | Target | Domain |
|---|---|---|---|---|---|
| AgeDB-DIR | 12,208 | 2,140 | 14,348 | Age (years) | In-the-wild faces |
| IMDB-WIKI-DIR | 81,911 | 11,016 | 92,927 | Age (years) | Web-scale faces |
| IMDB-Movie-DIR | 7,049 | 1,203 | 8,252 | IMDb movie score | Movie posters |
| BoneAge-DIR | 12,528 | 1,508 | 14,036 | Bone maturity (months) | Medical imaging |
| **Total** | **113,696** | **15,867** | **129,563** | – | – |

*Table 10.* Benchmarking results on AgeDB-DIR.

| Metrics | MSE ↓ | | | | MAE ↓ | | | | GM ↓ | | | |
|---|---|---|---|---|---|---|---|---|---|---|---|---|
| Shot | All | Many | Med. | Few | All | Many | Med. | Few | All | Many | Med. | Few |
| **Deep Imbalanced Regression** | | | | | | | | | | | | |
| VANILLA | 101.60 | 78.40 | 138.52 | 253.74 | 7.77 | 6.62 | 9.55 | 13.67 | 5.05 | 4.23 | 7.01 | 10.75 |
| DIR | 96.70 | 76.11 | 115.86 | 238.25 | 7.47 | 6.69 | 8.30 | 12.55 | 4.71 | 4.25 | 5.36 | 8.59 |
| RANKSIM | 82.10 | 68.60 | 102.61 | 152.84 | 6.91 | 6.34 | 7.79 | 9.89 | 4.28 | 3.92 | 4.88 | 6.89 |
| VIR | 81.76 | 70.61 | 91.47 | 142.36 | 6.99 | 6.39 | 7.47 | 9.51 | 4.41 | 4.07 | 5.05 | 6.23 |
| CONR | - | - | - | - | 6.81 | 6.32 | 7.45 | 9.21 | 4.39 | 3.81 | 5.01 | 6.02 |
| GROUP-DIR | - | - | - | - | 6.87 | 6.54 | 6.96 | 9.83 | 4.30 | 4.10 | 4.39 | 6.45 |
| HCA | - | - | - | - | 6.94 | 6.67 | 7.07 | 9.10 | - | - | - | - |
| DIST | - | - | - | - | 7.64 | 7.57 | 7.32 | 9.12 | 4.76 | 4.75 | 4.56 | 5.45 |
| **MLLM Regression—Qwen2.5-VL-3B** | | | | | | | | | | | | |
| ZEROSHOT | 342.12 | 416.72 | 156.29 | 147.83 | 13.40 | 15.56 | 8.24 | 7.22 | 5.24 | 7.16 | 2.18 | 3.08 |
| SFT | 69.45 | 59.19 | 88.58 | 113.41 | 6.37 | 5.78 | 7.67 | 8.36 | 1.86 | 1.48 | 3.19 | 3.75 |
| SFT-SOFT | 69.30 | 59.08 | 88.16 | 113.60 | 6.38 | 5.80 | 7.67 | 8.36 | 1.94 | 1.57 | 3.10 | 3.75 |
| VISUALQUALITY | 142.48 | 146.84 | 152.98 | 72.03 | 9.35 | 9.31 | 10.42 | 6.83 | 3.71 | 3.50 | 4.95 | **2.92** |
| REGRESSION REWARD | 57.47 | 51.67 | 66.84 | 86.62 | 5.85 | 5.48 | 6.52 | 7.58 | 1.88 | 1.52 | 3.01 | 3.70 |
| DISCO MAE REWARD | 59.53 | 54.91 | 70.83 | 71.85 | 5.95 | 5.64 | 6.73 | 6.75 | 1.84 | 1.59 | 2.53 | 3.00 |
| OURS | **52.58** | **51.30** | **52.51** | **65.0** | **5.52** | **5.42** | **5.62** | **6.40** | **1.63** | **1.43** | **2.07** | 3.04 |
| OURS VS. SFT | +24.29% | +13.33% | +40.72% | +42.69% | +13.34% | +6.23% | +26.73% | +23.44% | +12.37% | +3.38% | +35.11% | +18.93% |
| **MLLM Regression—Qwen2.5-VL-7B** | | | | | | | | | | | | |
| ZEROSHOT | 83.14 | 81.68 | 81.45 | 101.70 | 7.16 | 7.05 | 7.17 | 8.23 | 2.61 | 2.42 | 3.05 | 3.42 |
| SFT | 59.63 | 52.26 | 76.68 | 81.20 | 5.82 | 5.35 | 7.03 | 6.98 | 1.59 | 1.18 | 3.57 | 2.73 |
| SFT-SOFT | 59.85 | 52.47 | 77.12 | 81.81 | 5.82 | 5.35 | 7.03 | 6.98 | 1.53 | **1.12** | 3.66 | 2.72 |
| VISUALQUALITY | 109.04 | 107.26 | 126.81 | 76.74 | 8.11 | 7.95 | 9.05 | 7.02 | 3.12 | 3.06 | 3.86 | 2.17 |
| REGRESSION REWARD | 54.92 | 51.88 | 61.72 | 64.83 | 5.62 | 5.41 | 6.12 | 6.22 | 1.67 | 1.45 | 2.25 | 2.92 |
| DISCO MAE REWARD | 55.05 | 51.88 | 62.87 | 63.32 | 5.64 | 5.44 | 6.13 | 6.16 | 1.66 | 1.45 | 2.22 | 2.66 |
| OURS | **49.99** | **49.23** | **49.97** | **57.22** | **5.33** | **5.27** | **5.42** | **5.69** | **1.50** | 1.31 | **2.10** | **2.01** |
| OURS VS. SFT | +16.17% | +5.80% | +34.83% | +29.53% | +8.42% | +1.50% | +22.90% | +18.48% | +5.66% | -11.02% | +41.18% | +26.37% |

## B.4. BoneAge-DIR

BoneAge-DIR is constructed from the RSNA Pediatric Bone Age dataset (Halabi et al., 2019), consisting of pediatric hand radiographs annotated with skeletal age in months (0–228, at 1-month resolution). We preserve the naturally long-tailed training distribution and construct a balanced test set across skeletal age bins for fair evaluation. The resulting dataset contains 12,528 training images and 1,508 test images, with pronounced imbalance across age bins.

## B.5. Input–Output Formatting and Prompt Templates

To isolate the effect of learning objectives, all models are trained and evaluated under a *pure numeric prediction* setting: no chain-of-thought reasoning, explanatory text, or intermediate steps are allowed. Each task is formulated as a single-turn instruction that requires the model to output only a numeric value.

**Age Estimation Prompt (AgeDB-DIR / IMDB-WIKI-DIR)**

<image> Age estimation: How old is the person in the image? Please answer with only a number.

**Movie Rating Prediction Prompt (IMDB-Movie-DIR)**

*Table 11.* Benchmarking results on IMDB-Movie-DIR.

| Metrics | MSE ↓ | | | | MAE ↓ | | | | GM ↓ | | | |
|---|---|---|---|---|---|---|---|---|---|---|---|---|
| Shot | All | Many | Med. | Few | All | Many | Med. | Few | All | Many | Med. | Few |
| **MLLM Regression—Qwen2.5-VL-3B** | | | | | | | | | | | | |
| ZEROSHOT | 274.52 | 270.46 | 249.30 | 434.29 | 13.63 | 14.17 | 10.61 | 18.99 | 7.24 | 9.66 | **2.47** | 16.81 |
| SFT | 101.30 | **37.19** | 166.64 | 558.83 | 7.44 | **4.87** | 11.21 | 21.51 | 2.57 | **1.52** | 7.02 | 18.28 |
| SFT-SOFT | 101.36 | 37.28 | 165.67 | 562.45 | 7.43 | 4.87 | 11.15 | 21.57 | 2.52 | 1.52 | 6.60 | 18.33 |
| VISUALQUALITY | 857.38 | 879.06 | 866.66 | 549.30 | 24.43 | 24.44 | 25.25 | 20.23 | 14.75 | 13.71 | 18.44 | 13.42 |
| REGRESSION REWARD | 98.59 | 40.22 | 149.76 | 534.65 | 7.42 | 5.06 | 10.51 | 21.14 | 3.13 | 1.96 | 7.46 | 18.68 |
| DISCO MAE REWARD | 102.30 | 45.62 | 153.73 | 529.55 | 7.53 | 5.36 | 10.38 | 20.61 | 3.02 | 2.05 | 5.91 | 16.89 |
| OURS | **84.41** | 51.12 | **110.70** | **349.96** | **6.89** | 5.60 | **8.12** | **16.35** | **2.11** | 1.63 | 3.24 | **7.17** |
| OURS VS. SFT | +16.67% | -37.45% | +33.57% | +37.38% | +7.39% | -14.98% | +27.56% | +23.99% | +17.90% | -7.24% | +53.85% | +60.78% |
| **MLLM Regression—Qwen2.5-VL-7B** | | | | | | | | | | | | |
| ZEROSHOT | 1443.32 | 549.40 | 4318.98 | 514.16 | 10.42 | 7.18 | 17.66 | 18.91 | 3.43 | 2.48 | 7.72 | **5.86** |
| SFT | 76.63 | **31.53** | 123.98 | 392.81 | 6.42 | **4.36** | 9.58 | 17.14 | 2.01 | 1.21 | 5.69 | 10.88 |
| SFT-SOFT | 76.64 | 31.71 | 123.65 | 392.35 | 6.43 | 4.38 | 9.56 | 17.16 | 1.93 | 1.14 | 5.66 | 11.04 |
| VISUALQUALITY | 942.81 | 917.35 | 1083.67 | 697.31 | 24.73 | 23.38 | 28.88 | 24.04 | 12.35 | 10.30 | 19.24 | 17.31 |
| REGRESSION REWARD | 73.56 | 31.96 | 107.53 | 401.18 | 6.23 | 4.36 | 8.64 | 17.73 | 1.69 | **0.95** | 5.12 | 13.90 |
| DISCO MAE REWARD | 78.78 | 33.32 | 117.79 | 429.83 | 6.46 | 4.46 | 9.14 | 18.35 | 1.89 | 1.07 | 5.77 | 14.71 |
| OURS | **66.86** | 39.71 | **82.07** | **306.53** | **5.95** | 4.85 | **6.87** | **14.58** | **1.62** | 1.33 | **1.92** | 7.20 |
| OURS VS. SFT | +12.75% | -25.94% | +33.80% | +21.96% | +7.32% | -11.24% | +28.29% | +14.94% | +19.40% | -9.92% | +66.26% | +33.82% |

<image> You are given a movie poster. Using only the visual cues in the poster, predict the movie's IMDb rating score as accurately as possible. Return only one integer between 0 and 100 (IMDb score × 10).

**Bone Age Estimation Prompt (BoneAge-DIR)**

<image> You are given a pediatric hand radiograph. Please assess the skeletal age based on the image. Task: Bone age estimation. Definition: Skeletal age is the estimated developmental age of the bones, measured in months. Constraints: - Minimum value: 0 months. - Maximum value: 216 months. - Step value: 1 month. Question: What is the skeletal age (in months) shown in this radiograph? Output a single integer number only.

**Unified Answer Template**  For all tasks and all training stages, we enforce a unified answer template to standardize output parsing and reward computation. Concretely, the question is appended with:

{Questions} Please output the final answer in <answer> </answer> tags.

This ensures robust numeric extraction and decouples reward computation from free-form text generation.

**Output Parsing and Reward Composition**

**Numeric Parsing.**  We extract the first numeric value enclosed by <answer> and </answer> tags using a regular expression. Outputs without a valid numeric value are treated as invalid.

**Format Reward.**  In addition to the CCC reward, we include a lightweight *format reward* to enforce valid outputs during RL. Malformed generations (missing tags, non-numeric strings, or out-of-range values) yield undefined or noisy reward signals; the format reward filters such cases and stabilizes training. Following our implementation, a valid output (correct tag format + parseable number + within the valid range) receives a small constant reward $c$ (set to $0.5$), and invalid outputs receive 0:

$$r_{\mathrm{fmt}} = \begin{cases} c, & \text{valid format and within range,} \\ 0, & \text{otherwise.} \end{cases}$$

**Final Reward.**  The overall reward used for GRPO optimization is:

$$r = r_{\mathrm{CCC}} + r_{\mathrm{fmt}}.$$

Once the model learns the output format, $r_{\mathrm{fmt}}$ becomes constant and optimization is dominated by the CCC-based batch-level supervision.

*Table 12.* Benchmarking results on IMDB-WIKI-DIR.

| Metrics | MSE ↓ | | | | MAE ↓ | | | | GM ↓ | | | |
|---|---|---|---|---|---|---|---|---|---|---|---|---|
| Shot | All | Many | Med. | Few | All | Many | Med. | Few | All | Many | Med. | Few |
| **Deep Imbalanced Regression** | | | | | | | | | | | | |
| VANILLA | 138.06 | 108.70 | 366.09 | 964.92 | 8.06 | 7.23 | 15.12 | 26.33 | 4.57 | 4.17 | 10.59 | 20.46 |
| DIR | 129.35 | 106.52 | 311.49 | 811.82 | 7.78 | 7.20 | 12.61 | 22.19 | 4.37 | 4.12 | 7.39 | 12.61 |
| RANKSIM | 123.18 | 100.86 | 280.55 | 879.85 | 7.35 | 6.81 | 11.50 | 22.75 | 4.05 | 3.85 | 6.05 | 14.68 |
| VIR | 118.94 | 96.10 | 295.79 | 771.47 | 7.19 | 6.56 | 11.81 | 20.96 | 3.85 | 3.63 | 6.51 | 12.23 |
| CONR | - | - | - | - | 7.33 | 6.69 | 11.87 | 21.53 | 3.99 | 3.81 | 6.66 | 12.62 |
| GROUP-DIR | - | - | - | - | 7.22 | 6.71 | 11.42 | 20.25 | 3.88 | 3.68 | 5.74 | 11.13 |
| HCA | - | - | - | - | 7.31 | 6.82 | 10.99 | 22.04 | - | - | - | - |
| DIST | - | - | - | - | 8.02 | 7.48 | 12.30 | 22.33 | 4.59 | 4.37 | 7.08 | 13.02 |
| **MLLM Regression—Qwen2.5-VL-3B** | | | | | | | | | | | | |
| ZEROSHOT | 182.82 | 155.86 | 286.83 | 1785.71 | 9.44 | 9.09 | 11.59 | 24.09 | 3.46 | 3.38 | 3.84 | 10.92 |
| SFT | 119.14 | 98.28 | 278.37 | 764.64 | 7.10 | 6.53 | 11.73 | 21.88 | 1.81 | 1.60 | 5.89 | 11.84 |
| SFT-SOFT | 118.96 | 98.11 | 278.15 | 762.90 | 7.10 | 6.53 | 11.72 | 21.88 | 1.84 | 1.63 | 5.88 | 11.88 |
| VISUALQUALITY | 389.98 | 381.66 | 430.66 | 815.35 | 16.12 | 15.99 | 17.13 | 20.36 | 9.12 | 8.84 | 13.02 | **9.18** |
| REGRESSION REWARD | 120.42 | 97.12 | 287.01 | 891.66 | 7.17 | 6.60 | 11.68 | 23.06 | 1.92 | 1.77 | 3.77 | 13.26 |
| DISCO MAE REWARD | 119.67 | 100.39 | 264.49 | 736.0 | 7.13 | 6.66 | 10.88 | 20.78 | 1.83 | 1.68 | 3.91 | 9.94 |
| OURS | **112.50** | **92.51** | **250.13** | **704.80** | **6.71** | **6.26** | **9.89** | 20.17 | **1.60** | **1.53** | **2.06** | 12.35 |
| **OURS VS. SFT** | +5.57% | +5.87% | +10.14% | +7.83% | +5.49% | +4.13% | +15.69% | +7.82% | +11.60% | +4.38% | +65.03% | -4.31% |
| **MLLM Regression—Qwen2.5-VL-7B** | | | | | | | | | | | | |
| ZEROSHOT | 123.37 | 103.52 | 263.67 | 832.58 | 7.42 | 6.95 | 10.90 | 23.06 | 2.11 | 1.98 | 3.38 | 16.53 |
| SFT | 110.20 | 90.74 | 255.12 | 742.25 | 6.61 | 6.11 | 10.56 | 20.44 | 1.46 | 1.32 | 3.65 | 9.86 |
| SFT-SOFT | 110.15 | 90.79 | 253.93 | 741.58 | 6.61 | 6.11 | 10.52 | 20.44 | 1.44 | 1.30 | 3.47 | 9.91 |
| VISUALQUALITY | 148.76 | 133.97 | 246.81 | 731.62 | 8.56 | 8.33 | **10.01** | **18.29** | 2.96 | 2.90 | 3.42 | **5.40** |
| REGRESSION REWARD | 110.30 | 91.01 | 255.73 | 721.38 | 6.62 | 6.13 | 10.51 | 20.64 | 1.45 | 1.34 | 2.79 | 12.03 |
| DISCO MAE REWARD | 111.32 | 91.61 | 260.64 | 729.39 | 6.64 | 6.13 | 10.77 | 20.59 | 1.41 | 1.30 | 2.91 | 13.28 |
| OURS | **106.72** | **87.96** | **246.67** | **713.50** | **6.41** | **5.95** | 9.96 | 19.94 | **1.33** | **1.25** | **2.06** | 10.31 |
| **OURS VS. SFT** | +3.16% | +3.06% | +3.31% | +3.87% | +3.03% | +2.62% | +5.68% | +2.45% | +8.90% | +5.30% | +43.56% | -4.56% |

## C. Experimental Details

All experiments in the main paper are conducted by fine-tuning Qwen2.5-VL-3B/7B as the multimodal large language model backbone. We adopt Group Relative Policy Optimization (GRPO) for post-training, and apply LoRA adapters with rank 64 and scaling factor 128. Optimization is performed using AdamW with an initial learning rate of $1 \times 10^{-5}$ and linear decay. For GRPO, we use a batch size of 16, sample $K = 4$ candidate generations per input, and set the KL coefficient to $\beta = 0.04$ across all datasets. GRPO training is performed for 4 epochs on AgeDB-DIR, IMDB-Movie-DIR, and BoneAge-DIR, and for 2 epoch on IMDB-WIKI-DIR. For supervised fine-tuning (SFT) baselines, we use the same backbone and prompt format, with a batch size of 32 and identical LoRA configurations.; SFT is trained for 2 epochs on AgeDB-DIR, IMDB-Movie-DIR, and BoneAge-DIR, and for 1 epoch on IMDB-WIKI-DIR. Across all experiments, we ensure identical input formatting, numeric decoding, and evaluation protocols between SFT and GRPO.

**Classical Deep Imbalanced Regression Methods.** We include representative deep imbalanced regression (DIR) methods originally proposed for conventional visual regression models, including DIR (Yang et al., 2021), RankSim (Gong et al., 2022), VIR (Wang & Wang, 2023), ConR (Keramati et al.), Group-DIR (Pu et al., 2025), HCA (Xiong & Yao, 2024), and DIST (Nie et al.). These methods are designed for non-generative, CNN-based regression pipelines that predict continuous values with explicit regression heads, and are not designed for generative or autoregressive multimodal language models. We report their results as reference points to contextualize the performance gap between classical vision-based regression methods and generative MLLM-based numeric generation.

**Zero-Shot Baseline.** ZEROSHOT denotes direct numeric generation from the pretrained Qwen2.5-VL-3B/7B model without any task-specific fine-tuning. This setting reflects the inherent numeric prediction capability of the backbone under the given prompt and serves as a lower-bound reference.

*Table 13.* Benchmarking results on BoneAge-DIR.

| Metrics | MSE ↓ | | | | MAE ↓ | | | | GM ↓ | | | |
|---|---|---|---|---|---|---|---|---|---|---|---|---|
| Shot | All | Many | Med. | Few | All | Many | Med. | Few | All | Many | Med. | Few |
| **MLLM Regression—Qwen2.5-VL-3B** | | | | | | | | | | | | |
| ZeroShot | 13979.17 | 14441.79 | 15428.21 | 11031.82 | 95.36 | 112.61 | 82.90 | 79.98 | 61.44 | 104.10 | 59.52 | 19.14 |
| SFT | 575.72 | 488.79 | 591.27 | 747.42 | 18.60 | 17.53 | 18.22 | 21.86 | 5.16 | 2.56 | 7.85 | 11.39 |
| SFT-Soft | 581.07 | 492.28 | 599.22 | 751.91 | 18.70 | 17.59 | 18.39 | 21.91 | 5.23 | 2.63 | 7.94 | 11.38 |
| VisualQuality | 2291.83 | 2452.44 | 2460.24 | 1581.02 | 39.13 | 39.30 | 42.06 | 32.86 | 20.57 | 12.86 | 32.69 | 24.38 |
| Regression Reward | 399.97 | 366.63 | 391.54 | 494.54 | 15.45 | 14.87 | 15.14 | 17.42 | **3.86** | **2.13** | 4.79 | 9.99 |
| Disco MAE Reward | 762.26 | 686.10 | 741.07 | 982.20 | 20.87 | 21.07 | 19.31 | 23.51 | 6.21 | 5.54 | 5.29 | 11.16 |
| Ours | **346.45** | **339.09** | **353.06** | **350.43** | **14.22** | **14.15** | **14.22** | **14.35** | 4.74 | 4.17 | **4.30** | **7.80** |
| **Ours vs. SFT** | +39.82% | +30.63% | +40.29% | +53.11% | +23.55% | +19.28% | +21.95% | +34.35% | +8.14% | -62.89% | +45.22% | +31.52% |
| **MLLM Regression—Qwen2.5-VL-7B** | | | | | | | | | | | | |
| ZeroShot | 13644.19 | 14006.14 | 14676.44 | 10739.22 | 93.69 | 110.65 | 81.34 | 78.76 | 59.70 | 101.95 | 57.17 | 18.68 |
| SFT | 484.47 | 433.82 | 485.29 | 600.95 | 17.09 | 16.20 | 16.82 | 19.71 | 3.87 | 1.67 | 5.40 | 14.30 |
| SFT-Soft | 479.59 | 430.59 | 471.81 | 609.39 | 17.01 | 16.10 | 16.61 | 19.90 | 3.82 | **1.62** | 5.34 | 14.44 |
| VisualQuality | 2068.40 | 2073.97 | 2332.22 | 1528.71 | 37.78 | 36.94 | 41.14 | 33.04 | 19.38 | 11.53 | 30.99 | 25.50 |
| Regression Reward | 396.77 | 330.11 | 430.02 | 485.82 | 15.00 | 14.47 | 14.79 | 16.66 | 3.43 | 1.96 | 4.00 | 9.35 |
| Disco MAE Reward | 722.96 | 674.48 | 684.93 | 911.96 | 20.17 | 18.66 | 20.34 | 23.36 | 5.93 | 2.56 | 9.19 | 17.44 |
| Ours | **299.36** | **289.97** | **317.64** | **284.78** | **13.35** | **13.35** | **13.69** | **12.67** | **3.08** | 2.30 | **3.31** | **5.26** |
| **Ours vs. SFT** | +38.21% | +33.16% | +34.55% | +52.61% | +21.88% | +17.59% | +18.61% | +35.72% | +20.41% | -37.72% | +38.70% | +63.22% |

**Supervised Fine-Tuning Baseline.** SFT denotes standard supervised fine-tuning using token-level cross-entropy for autoregressive numeric generation.

**SFT-Soft** is a soft variant of SFT that introduces token-level reweighting to incorporate weak numeric distance awareness during training (Wang et al., 2025b). Specifically, we identify digit tokens corresponding to numeric outputs and assign larger loss weights to positions where the model's digit prediction deviates more from the ground-truth digit. The weight is proportional to the absolute difference between the predicted and target digits, with clipping applied for numerical stability, while all non-numeric tokens retain unit weight. This design preserves the standard autoregressive training objective while partially reflecting numeric distance at the token level, alleviating the brittleness of pure cross-entropy for regression-like targets. We note that SFT-Soft is a faithful re-implementation based on the method description in prior work (Wang et al., 2025b), as no official code release is available. All hyperparameters are fixed across datasets to ensure fair comparison. Both SFT and SFT-Soft use identical prompts and decoding strategies as GRPO-based methods, and differ only in the training objective.

**Reward-Based and RL Post-Training Baselines.** We further compare against several reinforcement learning and reward-based post-training strategies for numeric prediction in MLLMs, including VisualQuality (Wu et al., 2025), Standard Regression Reward, and DISCO MAE Reward (Zhou et al., 2025). These methods differ primarily in how reward signals are constructed and how supervision is propagated during policy optimization. We consider representative reinforcement learning reward formulations for numeric prediction in MLLMs.

**Standard Regression Reward** directly optimizes point-wise numeric accuracy using absolute error (e.g., MAE) as the reward signal. While simple and intuitive, this formulation is dominated by high-frequency regions under long-tailed label distributions and is prone to regression-to-the-mean behavior.

**DISCO MAE Reward** extends MAE-based regression rewards with frequency-aware reweighting. Following the DISCO framework (Zhou et al., 2025), we partition the training data into bins and rescale the reward signal based on bin prevalence, allowing rarer regions to exert stronger influence during optimization. In our implementation, we follow DISCO's reward scaling strategy and disable the standard GRPO variance normalization to preserve the absolute effect of frequency weights. While this approach partially alleviates head dominance, it remains a point-wise regression objective and does not explicitly model distributional structure across samples.

**Ranking- and Correlation-Based Rewards.** Ranking-based rewards, exemplified by VisualQuality-R1 (Liu et al., 2025b), formulate learning objectives through pairwise preference comparisons. These methods assume that predictions encode uncertainty and optimize relative ranking consistency across responses or samples. They have been shown effective for perceptual assessment and *ordinal regression* tasks with a limited label range (e.g., 0–5), where preserving relative order is the primary objective. In contrast, we study *general deep imbalanced regression* in MLLMs, where targets are continuous,

unbounded or wide-range, and follow highly skewed long-tailed distributions. In this setting, optimizing ordinal consistency alone is insufficient. Ranking-based rewards do not directly optimize absolute numeric accuracy, nor do they explicitly preserve regression-specific properties such as scale calibration and mean alignment, which are critical for reliable prediction across both dense and sparse target regions.

**Spearman Correlation Reward.** In addition to pairwise ranking rewards, we also consider correlation-based supervision via the Spearman correlation coefficient (Wissler, 1905). A Spearman correlation reward measures the monotonic consistency between predicted values and ground-truth targets within a batch, encouraging correct global ordering while remaining agnostic to absolute scale. In our setting, we implement a *batch-level Spearman reward* by computing the Spearman correlation between predictions and ground-truth labels across samples in the same minibatch and using it as the reinforcement learning signal. This formulation provides a strong ordering-aware baseline.

Despite their effectiveness in enforcing relative ordering, ranking-based and Spearman-based rewards do not explicitly constrain absolute scale or mean alignment. As a result, while they can substantially improve tail performance under long-tailed settings, they may fail to preserve accuracy in dense, many-shot regions where precise numeric calibration is critical. In particular, Spearman-based rewards do not suffer from regression-to-the-mean collapse. Instead, their limitation lies in the absence of absolute regression supervision, leading to degraded performance in high-density regions despite strong ordering consistency. This behavior is clearly reflected in Table 5, where batch-level Spearman rewards improve few-shot accuracy but underperform on many-shot samples.

**CCC Reward.** Importantly, Spearman-based rewards and our CCC-based reward are *not* conceptually conflicting. Both operate on batch-level comparisons and leverage cross-sample relational supervision. Our empirical results indicate that the primary driver of performance improvement under severe imbalance is the use of *batch-level comparison itself*, which is *entirely absent in SFT and point-wise MAE-based rewards*. Building on this insight, our method introduces a batch-level, distribution-aware *regression* reward that extends beyond pure ordering consistency. By jointly aligning correlation, scale, and mean across samples, the CCC-based reward provides a more complete supervision signal for long-tailed numeric prediction. Unlike ranking-based rewards, it directly optimizes continuous numeric structure; unlike MAE-based rewards (with or without reweighting), it captures global distributional relationships beyond point-wise errors.

## D. Extended Discussion and Analysis

**Batch-Level Supervision for Long-Tailed Regression in MLLMs.** The core contribution of this work is to reformulate numeric prediction in MLLMs as a *batch-level, distribution-aware learning problem*, instead of optimizing isolated point-wise errors. Under long-tailed target distributions, point-wise objectives (either token-level CE in SFT, or value-level MAE/MSE rewards in RL) are dominated by the many-shot region and thus tend to produce regression-to-the-mean behavior. In contrast, our GRPO formulation enables *batch-relative supervision*, where each prediction is evaluated through its relation to other samples within the same minibatch, making the learning signal explicitly sensitive to distributional structure, rather than marginal accuracy alone, and naturally exposes tail samples to non-vanishing supervision without reweighting or resampling.

**Sensitivity to Batch Size and Batch Statistics.** Our method leverages minibatch-level statistics as a stochastic proxy for global distributional structure. While this introduces a dependency on batch size, empirical results indicate that performance improvements saturate with moderate batch sizes and a small number of sampled generations. Importantly, the approach does not rely on label-aware or stratified batching, and remains effective under standard random sampling, suggesting robustness to realistic training conditions. Nevertheless, we acknowledge that extremely small or highly non-representative batches may introduce noise in the reward signal, and deeper analysis of batch composition effects remains an important direction for future investigation.

**Choice of CCC and Generality of the Framework.** We instantiate the batch-level reward using the Concordance Correlation Coefficient (CCC) because it is bounded and numerically stable, and it jointly measures (i) correlation, (ii) scale consistency, and (iii) mean alignment between predicted and ground-truth values. Importantly, our framework is *not* specific to CCC. Any group-level objective that compares predicted values with ground-truth values at the *set* level can be used within the same GRPO-based optimization pipeline, including rank-based correlations (e.g., Spearman/Kendall), optimal-transport distances, or task-specific distributional measures. Our ablation results show that the main gains come from *batch-level supervision itself*, while CCC serves as an effective and simple instantiation that discourages variance collapse and mean shift under severe imbalance.

*Table 14.* Extension to 2D gaze estimation.

| Method | PitchMAE↓ | | | | YawMAE↓ | | | | AngularError↓ | | | |
|---|---|---|---|---|---|---|---|---|---|---|---|---|
| | All | Many | Med. | Few | All | Many | Med. | Few | All | Many | Med. | Few |
| PureRegression | 3.54 | 3.06 | 3.42 | 4.30 | 7.19 | 5.67 | 7.31 | 9.03 | 8.54 | 6.94 | 8.54 | 10.59 |
| CCC | 3.49 | 3.04 | 3.36 | 4.23 | 6.77 | 5.26 | 6.76 | 8.72 | 8.13 | 6.62 | 7.97 | 10.25 |
| CCC-Joint | 3.48 | 2.98 | 3.36 | 4.25 | 6.59 | 5.25 | 6.47 | 8.43 | 7.94 | 6.51 | 7.75 | 9.99 |

**Extension Beyond One-dimensional Regression.** In this work, we focus on scalar-valued regression tasks, which constitute a large class of practical MLLM applications (e.g., age estimation, medical scores, rating prediction) and allow for controlled analysis of long-tailed behavior. Importantly, our framework is not inherently restricted to one-dimensional outputs. The batch-level reward formulation operates on sets of predicted values and ground-truth values, and can be extended to low-dimensional continuous targets by applying CCC (or its multivariate variants) per dimension or via joint covariance alignment. To provide preliminary evidence beyond scalar regression, we additionally evaluate CCC-GRPO on a 2D gaze estimation task with pitch and yaw, as shown in Table 14. Concretely, we apply the CCC reward per dimension and combine it with an angular consistency term. The model architecture and training pipeline remain unchanged; only the reward computation is generalized from scalar moments to dimension-wise moments. The results show consistent improvements in pitch MAE, yaw MAE, and angular error, indicating that the proposed framework is not limited to 1D regression.

Our claims of generality are scoped to the learning principle rather than empirical coverage. Specifically, we claim that batch-level, distribution-aware rewards address a fundamental failure mode of point-wise supervision under long-tailed regression, which is independent of model architecture or modality. While we demonstrate this principle on several representative MLLM regression benchmarks, extending empirical validation to broader regression settings remains an important direction for future work rather than a prerequisite for the validity of the proposed formulation.

**Why Mean Predictions Are Used as Context.** We use the mean prediction of the same sample across multiple stochastic generations, denoted as $\{\mu(x_j)\}_{j \neq i}$, as a contextual anchor to provide a low-variance estimate of each sample's prediction distribution. Importantly, the primary role of other samples in our reward design is *not* to perform fine-grained pairwise comparison between individual generations, but to serve as a reference for estimating the *batch-level distributional structure*. Accordingly, the contextual signal should reflect the overall prediction distribution of the minibatch, rather than the stochastic variability of any single generation.

This design makes the reward for each sampled prediction $q_i^{(k)}$ sensitive to global distributional structure, while avoiding unstable cross-sample coupling among stochastic generations. In contrast, directly using all sampled predictions from other samples as context would significantly increase reward variance and computational complexity. Specifically, if each of the other $N-1$ samples has $K$ stochastic generations, then for a single sampled prediction $q_i^{(k)}$, the number of possible relational orderings scales as $K^{N-1}$. Such combinatorial explosion makes the relative ranking of a single generation highly sensitive to random sampling noise, especially when $K$ is small.

By aggregating predictions from other samples into their empirical means, we obtain a stable and low-variance contextual reference. This preserves distribution-level relational information while ensuring that reward computation remains stable, reproducible, and well-behaved under limited multi-generation sampling. Empirically, our results show that this mean-based reference is sufficient to suppress prediction collapse and improve tail reliability across diverse datasets.

**Computational Considerations.** Our method requires multi-generation sampling within GRPO during training. Empirically, performance saturates with a small number of generations (e.g., $K = 4$), and the method operates in a no-thinking regime without chain-of-thought or iterative reasoning. In practice, taking AgeDB-DIR as an example, GRPO training takes approximately 3 hours under our experimental setting, compared to around 30 minutes for supervised fine-tuning with the same backbone and LoRA configuration. This additional cost reflects the inherent overhead of reinforcement learning with multiple sampled trajectories. While CCC-GRPO trades increased training time for improved robustness under long-tailed distributions, improving the efficiency of RL-based post-training—such as reducing sampling overhead or accelerating convergence—remains an important direction for future work.

# E. Why Batch-Level CCC Mitigates Long-Tailed Regression Collapse

We provide an intuitive analysis of why batch-level concordance-based rewards mitigate regression collapse under long-tailed distributions.

**Limitations of Point-wise Objectives.** Let $p_{\text{train}}(y)$ be the imbalanced training distribution. Point-wise regression (e.g., minimizing $\mathbb{E}_{(x,y)\sim p_{\text{train}}} |f(x) - y|$) is dominated by the many-shot region, making predictions biased toward high-density values. As imbalance increases, a predictor that concentrates outputs around the head region can achieve low average error while performing poorly on tail values, producing the classic regression-to-the-mean failure mode. This effect is further amplified in MLLMs, where continuous values are generated autoregressively via discrete tokens, weakening value-level supervision under severe imbalance.

**CCC Penalizes Collapse via Covariance, Variance, and Mean Alignment.** Given a set of predicted values $\mathbf{q}$ and targets $\mathbf{y}$, CCC is

$$\text{CCC}(\mathbf{q}, \mathbf{y}) = \frac{2\,\text{Cov}(\mathbf{q}, \mathbf{y})}{\text{Var}(\mathbf{q}) + \text{Var}(\mathbf{y}) + (\mu_{\mathbf{q}} - \mu_{\mathbf{y}})^2}.$$

A degenerate predictor that outputs a constant value yields $\text{Var}(\mathbf{q}) \approx 0$ and $\text{Cov}(\mathbf{q}, \mathbf{y}) \approx 0$, which drives CCC toward zero regardless of how close the constant is to the global mean. Moreover, CCC explicitly penalizes mean shift and scale mismatch, preventing solutions that preserve ordering but distort magnitude. This distinguishes CCC from pure rank-based objectives, which preserve ordering but remain agnostic to absolute scale and mean, and are therefore insufficient for continuous regression tasks where magnitude matters.

**Why Batch-Level Comparison Matters.** In our reward construction, each sampled prediction is evaluated relative to other samples in the minibatch, rather than in isolation. For a tail sample, collapsing toward the head region simultaneously reduces covariance with batch targets and increases mean mismatch in the comparison set, leading to lower CCC rewards. This introduces a tail-sensitive learning signal *without* explicit reweighting or resampling. Importantly, the batch-level comparison acts as a stochastic proxy for global distributional alignment: each minibatch provides a local but unbiased estimate of relational structure, enabling scalable optimization without requiring full-dataset statistics. Although CCC is computed on minibatch-level statistics, GRPO optimizes relative advantages within each group, making learning driven by comparative ranking rather than absolute reward magnitude. This substantially mitigates variance induced by noisy batch estimates. While this discussion is not a formal proof, it explains why batch-level CCC rewards are well aligned with the failure modes of long-tailed regression in MLLMs.

