# OpenReview forum: "Injecting Distributional Awareness into MLLMs via Reinforcement Learning for Deep Imbalanced Regression"
_ICML.cc/2026/Conference — ICML 2026 regular_

### Official Review · Reviewer_rNoW · 2026-02-19

**Soundness:** 3
**Presentation:** 4
**Significance:** 3
**Originality:** 3
**Overall Recommendation:** 5
**Confidence:** 5

**Summary:**

The paper "Injecting Distributional Awareness into MLLMs via Reinforcement Learning for Deep Imbalanced Regression" identifies a critical flaw in the existing paradigm of multimodal large language models (MLLMs) in the context of numerical regression, especially in long-tailed and imbalanced distributions. The existing defective paradigm, which relies on token-wise cross-entropy or point-wise rewards, encourages the model to ignore numerical values and instead converge to the mean in areas of high sample density.

To solve this problem, a new framework of learning is proposed by the authors, changing the learning paradigm from individual samples to batches of samples. This framework is called CCC-GRPO and includes a reward function based on the Concordance Correlation Coefficient and a learning algorithm called Group Relative Policy Optimization. Unlike the existing paradigm, which checks whether the predicted numerical value is correct, the new paradigm forces the model to be consistent with the ground truth distribution in terms of correlation, scale, and mean. The framework is elegant in the sense that it does not require any architectural changes in the model.

Key Contributions
1.  Distributional Framework of DIR in MLLMs: In the context of multimodal large language models, the authors propose a distribution-aware framework of learning, which extends the existing paradigm of Deep Imbalanced Regression (DIR).
2.  Proposed CCC-Based Reward Function: In the existing paradigm, the model is encouraged to ignore numerical values and instead converge to the mean in areas of high sample density. In contrast, the new framework incorporates a relational reward function based on the Concordance Correlation Coefficient, which penalizes variance collapse and mean shift, the two primary defects in the existing paradigm.
3.  MLLM-DIR Benchmark: In the context of the new framework, the authors propose a benchmark of learning, which consists of four datasets, namely, AgeDB, IMDB-WIKI, IMDB-Movie, and BoneAge, and a total of 129k samples.
4.  Performance Improvements Using the Proposed Framework: In the context of the Qwen2.5-VL model, which consists of 3B and 7B parameters, the authors demonstrate the efficacy of the new framework in achieving better results in the medium and few-shot regimes.

**Compliance With Llm Reviewing Policy:**

Affirmed.

**Ethical Review Concerns:**

My biggest concern is the authors' explicit neglect of the broader societal and ethical implications of the research. The authors state in the impact section of the research article, “There are many potential societal consequences of our work, none of which we feel must be specifically highlighted here.” This is an ethically and scientifically unacceptable omission for a research article introducing a reinforcement learning framework to manipulate the predictive distributions of Multimodal Large Language Models (MLLMs) for human-centric and medical data. I have raised two issues of concern for the scientific community.

1. Discrimination, Bias, and Fairness in Biometrics (AgeDB & IMDB-WIKI): The empirical validation of the proposed method is based on the AgeDB and IMDB-WIKI datasets for facial age estimation. It is an established fact in the computer vision research community that models often exhibit disparate performance and fairness issues across racial, ethnic, and gender groups. Since the proposed CCC-GRPO method is designed to change the model's weights for “tail” instances to optimize an imbalanced regression metric, it is essential to consider the demographic composition of the “tail.” What if the “tail” is dominated by minority groups? Will the model's CCC objective amplify the bias in the imbalanced regression metric?

2. Clinical Safety and Potential Patient Harm (BoneAge-DIR): The authors have empirically validated the proposed method for the BoneAge-DIR dataset for clinical bone-age estimation based on pediatric hand radiographs. This is a very sensitive domain. Could reinforcement learning introduce prejudices to the model based on its forecasted comparability with the overall numerical distributions associated with all batches? What may happen if a model repeatedly underplays or overstates a child’s bone age for a given pediatric subpopulation? The potential ramifications of these discrepancies could be tremendous in terms of the development and treatment of pediatric individuals due to endocrine dysfunction.

**Ethical Review Flag:**

Flag this paper for an ethics review.

**Ethics Expertise Needed:**

["Discrimination / Bias / Fairness Concerns"]

**Final Justification:**

The paper demonstrated strong scientific merit and timely insights into MLLMs, reaffirming my initial evaluation.

**Key Questions For Authors:**

My concern for authors is that under extreme enough conditions, when there is a large enough percentage of tail items present in a mini-batch compared to many-shot items, the CCC defined on those subsets will still yield mathematically meaningful gradients for those tail items; this is particularly true as the batch size gets smaller relative to the number of samples being modeled. In addition, if I can verify with sound analytical methods that the CCC will isolate and properly penalize a tail bite and therefore not wash away the signal from the tail item when there are many-shot bite models from the prior minbatch, I will increase the score under the category of Soundness.

1. The ability of an evaluation framework to scale to multi-dimensional continuous space. However, the current trial for the evaluation phase has primarily consisted of limited one-dimensional scalar regression (e.g., age, IMDb) measurements. Continuous and spatial perceptions in the real world require a great deal of multi-dimensional (3D) spatial intelligence, which is typically represented in terms of bounding box coordinates (x-axis, y-axis, z-axis), robot joint angles (angle of movement around xyz), and 3D visual spaces (3D visual-space perception). The authors allude to the fact that the evaluation framework is not inherently restricted to 1-D outputs only; rather, it can be extended through Joint Covariate Alignment.

2. Did you perform any test trials to see whether there would/should be computational differences between $y$ as one-dimensional and the computational costs associated with making an $N$-dimensional vector?

In the evaluation sections, the authors addressed this issue and identified the most critical limitations inherent to autoregressive data models measuring continuous variables. If the authors can demonstrate their distributional awareness/ability scales out into the nth dimensional space, this would dramatically improve the authors' impact-based evaluations of their overall work from a domain-specific solution number to a broader continuous evaluation. The training dynamics of reinforcement learning (RL) include both format sensitivity in the initial phase of training and the reward structure for stabilizing the RL computation. A deterministically formatted (i.e., always following the same rule) and deterministically successfully completed respective $R_{fmt}$ and $R_{CCC}$.

3. What is heuristic optimum sensitivity?
The heuristic optimum sensitivity refers to the characteristics of the heuristic heuristically composing the heuristically optimally composing the format, rather than to describing how sensitive the performance of either the $R_{fmt}$ or $R_{CCC}$ is to the heuristic format $R^{T}_{fmt}$.

There are major concerned implications for authors in the Methodology Presentation: An adequate exposition of the initial phase of training will greatly clarify the methodology of your study and enhance its cognitive integrity. You will bolster the confidence that the proposed RL loop is stable if you can provide convincing evidence that the demonstration of single-digit format parametrization in the period preceding the enforcement of the CNN format will be successful.

**Limitations:**

No, the authors provide a general and condescending impact statement that says, "There are many potential societal consequences of this work, none of which we feel must be specifically highlighted here." However, the empirical validation of the paper's results relies on the following areas, which are very closely associated with important societal and ethical implications that need to be addressed:

1. Demographic Bias in Age Estimation: Everyone in the field of machine learning knows that age estimation models have poor fairness, performing worse and having bias on different ethnicities and genders. Since your approach directly modifies the learned distributions of these targets to correct long-tailed imbalances, the authors should address the potential impact of the CCC reward on the error rates for the underrepresented demographics at the tail of the long-tailed distribution.

2. Clinical Safety in Medical Imaging: The area of bone age assessment is an important medical application in pediatric endocrinology, which can lead to the diagnosis of abnormal growth and genetic disorders in children. When an MLLM is fine-tuned to predict biological age from medical images, the prediction errors, especially the bias in the prediction, can lead to misdiagnosis and exacerbate the problem of health disparities.

3. Constructive Suggestion for Improvement: I would suggest that the authors rewrite the impact statement completely for the camera-ready version of the paper and address the following areas: How will the application of deep imbalanced regression techniques in MLLMs affect the algorithmic bias in the demographics of the target audience in the field of biometric systems? How will the application of the proposed generative regression techniques in medical images affect the algorithmic bias in the field of medical imaging, and how will the performance on the long-tailed medical image dataset be assessed not only in terms of the average absolute error but also in terms of fairness in the context of algorithmic bias in the field of medical imaging?

**Strengths And Weaknesses:**

(i). Soundness: The authors present strengths. First and foremost, the empirical validation of the proposed method is comprehensive, where the method has been evaluated using four different datasets, namely AgeDB, IMDB-WIKI, IMDB-Movie, and BoneAge, containing more than 129k samples in total. Furthermore, the experimental design employed in the evaluation process is sound, where the authors have performed a detailed ablation study, which effectively evaluates the impact of the CCC reward in a variety of generation counts and batch sizes. The authors’ approach in evaluating the limitations of the proposed method and analyzing in detail the degradation of the Geometric Mean (GM) metric in high-density regions for the BoneAge-DIR task, due to a metric-objective mismatch, is a reflection of scholarly integrity. The paper's weakness in the method heavily relies on empirical minibatch-level statistics, which are used as a stochastic substitute for global distributional structure. Although the authors have recognized that the use of extremely small and non-representative batches could lead to a noisy reward signal, they have not fully considered the impact of this on stability, especially in constrained environments.

(ii). Presentation: The authors present an easy-to-read structure while also integrating both types of simple single-variable regression analysis and more complex regression techniques to illustrate the variations between classical reasoning approaches and more advanced reasoning approaches. The diagrams provided for each training paradigm (e.g., SFT vs. pointwise GRPO vs. batch-level CCC-GRPO) clearly demonstrate the separate methodologies utilized to achieve similar outputs. While the authors recommend that their framework should allow for simple implementation using available development and analysis tools, the supplemental information indicates that there is also significant dependency on each implementation utilizing a particular template or format (i.e., using particular XML/HTML tags) that is then parsed numerically and computed to derive rewards. Including this formatting limitation into the main body of the text would help provide additional definition to assist with producing consistent outputs and achieving reproducible results.

(iii). Significance: The authors addressed the strengths in the current generative architectures, which are at a fundamental disadvantage because they cannot provide a direct mapping between the discrete level of token prediction and the continuous nature of numerical data. The proposed improvement in tail region regression accuracy provides a high degree of practical utility for applications in which data typically occur in naturally skewed distributions, such as projected medical imaging results or real-world age estimations. There is a weakness in the paper I am presenting for the author to catch. The framework is currently limited to 1-dimensional scalar regression problems. The potential extension to multivariate or structured continuous outputs has not yet been explored.

(iv). Originality: First, strengths in the paper Changing the focus away from isolated per-instance numerical error minimization toward batch-level relational integrity through the imposition of supervised learning targeting those relationships is a significant conceptual breakthrough in MLLM regression. Application of the Concordance Correlation Coefficient (CCC) to both correspondingly align correlation and scale and means via an RL loop is a very original synthesis of established statistical principles. Second, the weaknesses in the paper regarding the originality of this work reside exclusively in its definition of the problem and reward design; there is no novel method proposed herein, since the optimization algorithm used is the common GRPO method.

---

> ### Author Rebuttal · Authors · 2026-03-31
>
> ## On estimator variance under extreme imbalance
> We thank the reviewer for this important question. We confirm that no stratified or tail-aware sampling is used; all mini-batches are constructed via uniform random sampling. Our CCC reward is not a per-sample estimator driven by rare tail instances, but a *cross-sample structural signal* aligning mean, variance, and correlation across the batch.
>
> From a theoretical perspective, stability follows from boundedness and controlled influence. CCC is bounded in $[-1,1]$, preventing reward explosion. Under bounded targets (e.g., $[0,100]$), replacing any sample perturbs batch moments by at most $O(1/B)$, and since CCC is smooth:
> $$|\Delta \mathrm{CCC}| \le \frac{C}{B}, \quad \mathrm{Var}(\widehat{\mathrm{CCC}}_B)=O(1/B).$$
> Thus, even tail outliers cannot dominate the reward; the resulting noise is stochastic rather than adversarial.
>
> Optimization is further stabilized by GRPO (clipping + KL). Empirically, we observe consistent convergence without gradient explosion or collapse. Stability therefore arises from (i) bounded reward, (ii) $O(1/B)$ sample influence, and (iii) RL-level regularization.
>
> ## On number parsing and formatting dependence
> We agree this should be clarified. The structured format is only a parsing interface; the reward is defined on the *parsed scalar value*, not formatting tokens. To verify this, we remove the answer tag and report results in Table 1. Performance shows only minor variation, indicating robustness to formatting choice. Therefore, the method is not tied to XML/HTML-style tags; any deterministic parsing scheme suffices. We will move this clarification to the main text.
> |Method|MAE↓All|Many|Med.|Few|GM↓All|Many|Med.|Few|
> |---|---|---|---|---|---|---|---|---|
> |Ours|5.95|4.85|6.87|14.58|1.62|1.33|1.92|7.20|
> |Ours w/o Tag|6.08|4.98|7.05|14.45|1.59|1.36|1.70|6.53|
> |Ours w/o Tag1|5.90|4.78|6.96|14.22|1.91|1.46|3.29|4.80|
> ## On extension to multi-dimensional regression
> We have conducted an additional 2D regression experiment on gaze estimation (pitch and yaw). The CCC reward is applied per dimension and combined with an angular consistency term. The method consistently improves performance across all metrics, demonstrating that the framework naturally extends beyond 1D scalar regression without architectural changes.
> More generally, the extension to $d$-dimensional outputs only affects reward computation (moment statistics), while the model and training pipeline remain unchanged. We will include these results and discussion in the revised version.
> |Metrics|PitchMAE↓||||YawMAE↓||||AngularError↓||||
> |---|---:|---:|---:|---:|---:|---:|---:|---:|---:|---:|---:|---:|
> |Shot|All|Many|Med.|Few|All|Many|Med.|Few|All|Many|Med.|Few|
> |PureRegression|3.54|3.06|3.42|4.30|7.19|5.67|7.31|9.03|8.54|6.94|8.54|10.59|
> |CCC|3.49|3.04|3.36|4.23|6.77|5.26|6.76|8.72|8.13|6.62|7.97|10.25|
> |CCCJoint|3.48|2.98|3.36|4.25|6.59|5.25|6.47|8.43|7.94|6.51|7.75|9.99|
> ## On format sensitivity and pre-RL stability
> We clarify that RL is not initialized from a model without formatting ability. We start from an instruction-tuned model (Qwen2.5-VL-Instruct), which already produces structured numerical outputs in zero-shot settings. RL quickly stabilizes formatting via a lightweight format reward, after which training is dominated by the CCC reward. Thus, RL refines *numerical consistency and distributional alignment*, rather than learning formatting from scratch. The format is only a parsing interface; the core contribution lies in supervising complete numerical outputs and cross-sample structure.
>
> ## On safety and ethical considerations
> We thank the reviewer and agree the original impact statement was insufficient. We will revise it along two dimensions:
> #### (1) Fairness and demographic bias.
> In datasets such as AgeDB and IMDB-WIKI, long-tailed regions may correlate with underrepresented demographic groups. Optimizing distribution-level objectives does not guarantee fairness across subpopulations and may unintentionally shift error distributions. We will explicitly state that deployment requires subgroup-level evaluation, fairness auditing, and calibration analysis, rather than relying solely on aggregate metrics.
> #### (2) Clinical safety in BoneAge.
> In medical applications, reliability on common cases is critical. A purely distribution-level objective may introduce undesirable trade-offs if used without constraints. We emphasize that our method is a training-time technique, not a deployment prescription. In safety-critical settings, reward design should incorporate instance-level constraints (e.g., CCC + MAE hybrid), and evaluation must include clinically relevant subgroup analysis and failure-case auditing.
> We will revise the impact statement accordingly and explicitly clarify that fairness and safety evaluation are necessary prerequisites for any real-world deployment.

---

> > ### Author Rebuttal · Reviewer_rNoW · 2026-03-31
> >
> > The authors have produced a technical response that is well-designed and professional in nature that responds directly to the primary theoretical and empirical issues raised in my initial review of the paper.
> >
> > 1. Stability/Tail-Aware: The mathematical rationale using the Lipschitz-style stability argument ($O(1/B)$ sample influence), which demonstrates that gradient noise can be discounted and that “tail-washing” is not an issue with unevenly distributed mini-batches, confirms my concerns about the ability of the CCC reward to act as a stable and structurally consistent restriction instead of an unstable and variable estimator at the per-sample level.
> >
> > 2. Multi-Dimensional Scaling: The addition of new 2D gaze estimation results significantly adds to the foundation that the framework is not purely a "toy" and can be mathematically and practically extended to include multi-dimensional spatial tasks (pitch, yaw, and angle of error) without needing architectural changes, thereby directly answering an inquiry I had regarding the use of the framework to build intelligent spatial applications.
> >
> > 3. Ablation of Formatting Dependence: The finding from the Ours w/o Tag experiment demonstrates that the method is robust and not unduly dependent on a specific heuristic XML/HTML parsing technique.
> >
> > 4. Societal and Ethical Considerations: I applaud the authors' change in position and development of a thoughtful, comprehensive approach to addressing demographic bias (AgeDB) and clinical safety (BoneAge). The authors' commitment to rigorous subgroup-level evaluation and fairness audits for the deployment of these technologies in their real world will aid in correcting the camera-ready version prior to publication.
> >
> > 5. Final comments: Given the strength of the new empirical evidence (2D gaze estimation) and the clarity of the stability proof, I am now confident in the technical soundness and expanded significance of this work, so I will revise my score to accept this paper.
> >
> > Finally, additional review now that the paper has been moved to acceptance position, due to the author's significant address and improvement further.

---

> > > ### Author Response · Authors · 2026-04-01
> > >
> > > We sincerely thank the reviewer for the thoughtful follow-up and for recognizing the improvements made in the rebuttal.
> > >
> > > We are particularly encouraged that the additional analyses on estimator stability, the 2D gaze experiment, and the formatting ablations helped clarify the robustness and generality of the proposed framework.
> > >
> > > We also appreciate your recognition of the revised discussion on societal and ethical considerations. We fully agree that fairness, subgroup performance, and clinical safety are critical aspects, and we will further strengthen these discussions in the camera-ready version to ensure responsible positioning of the method.
> > >
> > > Thank you again for your constructive feedback and for your support of the work.

---

### Official Review · Reviewer_Hgzh · 2026-03-02

**Soundness:** 2
**Presentation:** 4
**Significance:** 3
**Originality:** 2
**Overall Recommendation:** 4
**Confidence:** 3

**Summary:**

This paper introduces a distribution-aware reinforcement learning framework to tackle Deep Imbalanced Regression (DIR) in Multimodal Large Language Models (MLLMs). The authors argue that standard token-level Supervised Fine-Tuning (SFT) leads to regression-to-the-mean behavior under long-tailed continuous targets. To mitigate this, they propose utilizing Group Relative Policy Optimization (GRPO) with a batch-level Concordance Correlation Coefficient (CCC) reward, aiming to align the predicted and ground-truth distributions in terms of correlation, scale, and mean. The method is evaluated on a newly constructed suite of four MLLM DIR benchmarks (including AgeDB-DIR and BoneAge-DIR), demonstrating improvements in medium- and few-shot regions over SFT and point-wise RL baselines.

**Compliance With Llm Reviewing Policy:**

Affirmed.

**Final Justification:**

Final Recommendation: Weak Accept

Final Justification:
This paper proposes a distribution-aware reinforcement learning framework (CCC-GRPO) to address Deep Imbalanced Regression (DIR) in MLLMs.

In my initial review (scored 3: Weak Reject), I raised three major concerns: the statistical variance of the micro-batch CCC estimator under long-tailed distributions, the lack of continuous-loss SFT baselines, and the severe clinical trade-offs observed on the BoneAge-DIR dataset.

The authors provided a rigorous and transparent rebuttal. The newly added continuous-SFT baselines (Table 1) convincingly demonstrated the limitations of token-level supervision, which effectively justifies the computational cost of the proposed multi-trajectory RL approach. Furthermore, the explanations regarding the bounded nature of the CCC estimator and GRPO clipping adequately addressed my concerns regarding gradient stability.

While the inherent clinical trade-off in high-density regions (as shown in the BoneAge dataset and hybrid reward experiments) remains a practical limitation, I appreciate the authors' honesty. Their commitment to explicitly scoping this method as a training-time benchmark rather than a deployment-ready system, and updating the Impact Statement with necessary safety caveats, is acceptable.

The rebuttal successfully resolved my primary methodological concerns. Accordingly, I am raising my score to a 4 and recommend a weak accept.

**Key Questions For Authors:**

1. Estimator Variance under Extreme Imbalance: Given a small batch size ($B \le 32$) and a highly skewed distribution, the probability of uniformly sampling a few-shot target is extremely low. When a tail sample is included, it acts as a statistical outlier in the CCC computation. How do you prevent the reward signal (and consequently the policy gradients) from experiencing extreme volatility? Did you employ any hidden stratified sampling strategies during batch construction?

2. Necessity of RL over Continuous SFT: Why didn't you compare CCC-GRPO against a more data-efficient SFT model trained with an expected-value continuous DIR loss (e.g., mapping token probabilities to an expected scalar and applying CCC or RankSim directly)? If such a continuous SFT approach achieves similar distributional alignment, what justifies the heavy $O(K)$ computational cost of RL sampling?

3. The Clinical Trade-off in BoneAge-DIR: The degradation of the GM metric in the many-shot region for the BoneAge dataset is concerning. In practical medical AI applications, miscalibrating common cases is a critical failure. Have you investigated hybrid reward functions (e.g., $r = \alpha \cdot \text{CCC} + (1-\alpha) \cdot \text{MAE}$) to explicitly bound instance-level errors in high-density regions while maintaining global structural alignment?

**Limitations:**

The authors have discussed the computational overhead of RL. However, the discussion of potential negative societal impacts is inadequate. Specifically, for tasks like medical age estimation (BoneAge-DIR), the methodology forces the model to align with a global long-tailed distribution, which inherently degrades performance on the densest part of the distribution. The authors must explicitly discuss the safety and ethical implications of deploying such "distribution-forcing" models in clinical settings, where instance-level reliability on common cases cannot be compromised.

**Strengths And Weaknesses:**

Strengths:

Problem Formulation: Identifying the semantic gap between discrete autoregressive token generation and continuous numerical topology in MLLMs is a critical and timely contribution.

Benchmark Contribution: The curation of a unified DIR benchmark for MLLMs, encompassing diverse domains including medical radiographs, provides a valuable resource for future research.

Weaknesses:

Soundness (Statistical Reliability of the Batch-Level Estimator): The core methodological contribution relies on computing the CCC over a mini-batch of size $B$ to act as a surrogate for global distributional alignment. However, the experiments utilize a remarkably small batch size ($B=16$ to $32$). In an extreme long-tailed distribution, a uniformly sampled mini-batch will almost exclusively consist of many-shot (head) samples. If a rare tail sample does appear, its leverage on the batch-level covariance $\text{Cov}(q, y)$ and variance $\text{Var}(q)$ will be extreme. This makes the reward signal $r_k(x_i)$ highly volatile and sensitive to sampling noise. The paper lacks a theoretical analysis or empirical justification of how policy gradients $\nabla_{\theta} J(\theta)$ remain stable without stratified sampling.

Soundness (Data-Efficiency and RL Overhead): The paper motivates the use of RL by highlighting the flaws of hard token-level cross-entropy. However, from a data-efficient learning perspective, a more computationally lightweight solution would be to apply classical continuous DIR losses directly to the expected numerical value of the predicted tokens $\mathbb{E}[y|x]$ during SFT. The absence of a rigorously designed continuous-loss SFT baseline makes the necessity of introducing the massive computational overhead of multi-generation RL (generating $K=4$ trajectories per sample) unconvincing.

Significance & Clinical Trade-off in Medical Domains (BoneAge-DIR): The evaluation on the BoneAge-DIR dataset reveals a severe practical limitation. The Geometric Mean (GM) error significantly worsens in the many-shot region compared to the SFT baseline. From a medical AI perspective, degrading the predictive accuracy on the most common, typical patient cases (many-shot) to force overall structural alignment is clinically unacceptable. A model that trades high reliability on common cases for marginal gains on rare cases has limited practical deployability in safety-critical domains.

---

> ### Author Rebuttal · Authors · 2026-03-31
>
> ## Statistical reliability of the batch-level estimator
> We confirm that no stratified sampling or special batch construction is used; all mini-batches are formed via uniform random sampling. We agree that estimator variance under extreme imbalance is an important concern. Despite this, we observe stable training across experiments due to the following factors.
> #### (1) Bounded reward. CCC is bounded in $[-1,1]$, preventing numerical explosion even with outliers.
> #### (2) Controlled single-sample influence. Under bounded targets, the effect of any sample on batch statistics scales as $O(1/B)$, so even tail samples cannot dominate the reward.
> #### (3) Cross-sample correction is not tail-only. CCC provides a structural signal aligning mean, variance, and correlation across *all regions* (many, medium, few), mitigating regression-to-the-mean. Thus, composition changes introduce stochastic noise rather than systematic bias.
> #### (4) RL-level stabilization. GRPO (clipping + KL) further constrains updates under noisy rewards. Together, CCC acts as a bounded low-variance estimator (variance $O(1/B)$), yielding stable gradients without stratified sampling.
>
> ## On the necessity of RL over continuous SFT
> We thank the reviewer for suggesting continuous-SFT alternatives. We implement strong baselines (Table 1), including expectation-based supervision, reweighting, focal losses, and soft-label methods. However, these remain token-level: supervision is decomposed over tokens and cannot enforce consistency over complete numerical predictions or across samples. In contrast, our RL formulation enables (i) optimization over decoded outputs, (ii) cross-sample relational supervision via batch rewards, and (iii) alignment of distributional properties (mean, variance, correlation), which are not expressible in token-level likelihood objectives. Empirically, these SFT variants underperform, especially in medium- and few-shot regions.
> |Method|MAE↓All|Many|Med.|Few|GM↓All|Many|Med.|Few|
> |---|---|---|---|---|---|---|---|---|
> |SFT+MAE|6.50|5.82|8.05|8.63|2.03|1.56|3.84|4.20|
> |SFT+Focal|6.49|5.81|8.07|8.57|2.06|1.59|3.85|4.22|
> |SFT+ComWeight|6.50|5.83|8.00|8.66|2.01|1.55|3.81|3.90|
> |SFT+JointSoft|6.49|5.81|8.05|8.59|1.99|1.52|3.98|3.86|
> |SFT+DecodeWeight|6.51|5.83|8.04|8.72|2.03|1.56|3.85|3.95|
> |SFT+SingleWeight|6.38|5.80|7.67|8.36|1.94|1.57|3.10|3.75|
> |**Ours**|**5.52**|**5.42**|**5.62**|**6.40**|**1.63**|**1.43**|**2.07**|**3.04**|
> ## On the clinical trade-off in BoneAge-DIR
> We agree that improvements in tail regions must not degrade common cases. Our goal in including BoneAge-DIR is *not* deployment, but to test generality across domains under long-tailed distributions. Thus, the observed trade-off reflects an *objective-design effect* rather than a limitation of the framework. We further evaluate hybrid rewards (CCC+MAE, Table 2) and find that simple linear combinations do not consistently improve performance and may slightly degrade results, indicating conflicting optimization signals. This does not imply MAE is unimportant, but that combining instance- and distribution-level objectives requires more principled designs (e.g., normalized rewards). For real-world deployment, stricter instance-level guarantees and task-specific design are required, which we will clarify.
> |Method|MAE↓All|Many|Med.|Few|GM↓All|Many|Med.|Few|
> |---|---|---|---|---|---|---|---|---|
> |CCC|13.35|13.35|13.69|12.67|3.08|2.30|3.31|5.26|
> |CCC+0.3MAE|13.88|13.50|14.19|14.18|2.29|0.92|3.40|8.76|
> |CCC+0.5MAE|14.42|14.04|14.56|15.02|3.21|1.54|4.68|8.44|
> |CCC+0.7MAE|14.24|13.72|14.47|15.01|3.12|1.60|4.29|7.81|
> |CCC+1MAE|14.35|13.93|14.67|14.67|2.94|1.26|4.94|7.50|
> |CCC+2MAE|14.36|14.00|14.54|14.86|3.13|1.46|4.34|9.66|
> ## On safety and ethical implications
> We agree that distribution-level objectives require caution in safety-critical domains. Our method is a *training-time strategy*, not a deployment prescription. In practice (e.g., BoneAge), safeguards are needed, including (i) hybrid rewards to bound instance-level errors, (ii) subgroup evaluation and fairness analysis, and (iii) calibration assessment in high-density regions. We will revise the impact statement accordingly.

---

> > ### Author Rebuttal · Reviewer_Hgzh · 2026-04-03
> >
> > I have read the rebuttal and the authors have effectively addressed my main concerns.
> >
> > The added continuous-SFT baselines (Table 1) clearly justify the necessity of the RL approach, and the clarifications on the CCC estimator's stability are sound. While Table 2 confirms the inherent clinical trade-off in dense regions (BoneAge), the authors are transparent about it. Given their commitment to explicitly framing this as a training-time benchmark and updating the Impact Statement with proper safety caveats, I find this acceptable.
> >
> > All major issues are resolved. I will raise my score accordingly.

---

> > > ### Author Response · Authors · 2026-04-03
> > >
> > > Thank you very much for your thoughtful review and for taking the time to carefully assess our rebuttal. We sincerely appreciate your positive feedback.
> > >
> > > We are glad that our additional experiments and clarifications, including the continuous-SFT baselines and the analysis of the CCC estimator, have addressed your concerns.
> > > We will further refine the manuscript to clearly present the clinical trade-offs and ensure the impact statement appropriately reflects the safety considerations.
> > >
> > > Thank you again for your support.

---

### Official Review · Reviewer_ZzCn · 2026-03-09

**Soundness:** 2
**Presentation:** 2
**Significance:** 2
**Originality:** 2
**Overall Recommendation:** 2
**Confidence:** 4

**Summary:**

This paper aims to address the deep imbalanced regression (DIR) problem, replacing traditional point-wise rewards with batch-level supervision in GRPO. By utilizing the Concordance Correlation Coefficient, it explicitly aligns the predicted and ground-truth distributions in terms of correlation, scale, and mean, effectively mitigating regression-to-the-mean collapse under long-tailed data distributions.

**Compliance With Llm Reviewing Policy:**

Affirmed.

**Final Justification:**

The authors' additional results in the rebuttal differ significantly from Table 7 in the original manuscripts.
Although the authors stated that this was due to seed selection issues, **the change in GM's result on the Few split from 4.43 in the manuscripts to 2.9 $\pm$  0.86 in the rebuttal** is clearly unreasonable in my opinion, and it also contradicts the authors' claim of stable training.

Thus, I believe this paper is not currently ready for publication. I suggest rejection.

**Key Questions For Authors:**

See weakness

**Limitations:**

See weakness

**Strengths And Weaknesses:**

## Strengths

1. This paper has a good motivation, explicitly formulating the deep imbalanced regression problem in MLLMs as a distribution-aware reinforcement learning problem.

2. The paper conduct many experiments to validate its proposed CCC-GRPO.

## Weakness

1. The proposed method offers only incremental novelty built upon GRPO. Relying on multiple samples to fit the data distribution is already a common strategy in imbalanced problems. Beisdes, the contribution lacks specific, fundamental algorithmic advancements tailored to the GRPO framework.

2. In Table 6, the authors evaluate the CCC reward across different GRPO variants. However, the paper fails to report the original performance of these variants without the CCC reward. Including these baselines is necessary to fully substantiate the claim that the improvements stem primarily from the proposed reward design rather than the underlying RL algorithms.

3. In my opinion, the proposed batch-level  methods require larger batch sizes to better capture the sample distribution, as exhibited in Figure 2. Yet, Table 7 shows that increasing the batch size to 32 leads to the poorest overall results. There is a clear performance trade-off where the "Few" region improves, but the "Many" region drops. I am deeply concerned about why larger batch sizes negatively impact the head region and fail to improve the overall metrics.

4. Furthermore, given the massive parameter scale of large models, scaling up the batch size is computationally highly demanding and memory-intensive. This constraint imposes significant practical limitations on the applicability and scalability of the proposed batch-dependent method.

---

> ### Author Rebuttal · Authors · 2026-03-31
>
> ## On novelty and relation to prior imbalanced learning
> We respectfully disagree with the assessment that our work is an incremental extension of GRPO or classical imbalanced learning. Our focus is fundamentally different: we address a limitation in *numerical understanding of MLLMs*, rather than improving conventional DIR benchmarks. In classical DIR, models operate on continuous outputs. In MLLMs, numerical values are generated as sequences of discrete tokens, leading to a *token-level decomposition issue*: cross-entropy treats digits independently and fails to preserve complete numerical semantics. To the best of our knowledge, this issue has not been systematically studied.
>
> Building on this, our contributions are three-fold:
>
> **(1) Problem formulation and benchmark.** We formulate *DIR for MLLMs* and introduce a unified benchmark across domains.
>
> **(2) Analysis of existing paradigms.** We show that both SFT and standard RL (e.g., point-wise GRPO) are fundamentally limited, as they operate at the token or per-sample level and cannot enforce consistency over complete numerical predictions or across samples.
>
> **(3) New supervision paradigm.** We propose *cross-sample supervision over complete predictions* via a CCC-based reward, shifting to batch-level structural alignment and enabling distribution-level supervision.
>
> While cross-sample signals exist in classical DIR, extending them to generative MLLMs requires operating on decoded predictions and relational structure across samples. We therefore argue that our contribution should be evaluated from the perspective of *generative MLLM supervision*, rather than as a variant of existing DIR methods.
> This distinction is critical, as evaluating the method purely through the lens of classical DIR overlooks the unique challenges introduced by token-based generative modeling in MLLMs.
>
> ## On the performance of RL variants without CCC reward
> We thank the reviewer for pointing out the missing baselines. We have added RL variants without CCC in Table 1. Across all variants (GRPO, RegGRPO, DrGRPO), CCC consistently improves both MAE and GM. Since the optimizer is unchanged, this isolates the effect of the reward design, showing gains come from CCC-based cross-sample supervision rather than the RL algorithm.
> |Metrics|MAE↓||||GM↓||||
> |---|---:|---:|---:|---:|---:|---:|---:|---:|
> |Method|All|Many|Med.|Few|All|Many|Med.|Few|
> |DrGRPO|6.08|5.65|6.99|7.72|1.97|1.56|3.52|3.51|
> |DrGRPO+CCC|5.73|5.59|5.97|6.36|1.58|1.35|2.05|3.33|
> |RegGRPO|6.00|5.53|7.10|7.44|1.91|1.61|2.61|3.87|
> |RegGRPO+CCC|5.56|5.41|5.81|6.25|1.59|1.51|1.60|2.65|
> |GRPO|5.85|5.48|6.52|7.58|1.88|1.52|3.01|3.70|
> |GRPO+CCC|5.52|5.42|5.62|6.40|1.63|1.43|2.07|3.04|
>
> ## On batch-size sensitivity and head--tail trade-off
> We agree that batch size influences the quality of the batch-level estimate and can induce a head--tail trade-off. However, this does not indicate instability. We have now repeated our experiments across five random seeds and report mean$\pm$std for batch size ablation study using AgeDB. As shown in Table 2, MAE varies only marginally, indicating stable optimization across batch sizes. Additional seed experiments show that MAE remains stable across runs, confirming robustness to stochastic batch composition. We interpret this as a controllable shift in optimization emphasis (head vs. tail), rather than a limitation.
> |Metrics|MAE↓||||GM↓||||
> |---|---:|---:|---:|---:|---:|---:|---:|---:|
> |Batch|All|Many|Med.|Few|All|Many|Med.|Few|
> |24|5.56±0.02|5.46±0.04|5.68±0.15|6.13±0.16|1.50±0.08|1.35±0.10|1.71±0.19|2.90±0.86|
> |28|5.58±0.03|5.44±0.07|5.77±0.10|6.30±0.19|1.58±0.08|1.44±0.10|1.73±0.18|3.13±0.42|
> |32|5.58±0.02|5.57±0.04|5.46±0.16|5.97±0.18|1.63±0.08|1.56±0.11|1.58±0.33|3.03±0.40|
>
> ## On scalability and practical applicability
> We agree that computational cost is important, but our method does not rely on large batch sizes. Unlike contrastive learning, the batch serves as a *moment estimator*, affecting variance rather than correctness. Empirically, moderate batch sizes already yield stable improvements. Our goal is not to replace lightweight DIR pipelines, but to address a fundamental limitation of token-based numerical prediction in MLLMs. From this perspective, CCC-GRPO is a post-training method for improving numerical consistency, where moderate overhead is justified.

---

> > ### Author Rebuttal · Reviewer_ZzCn · 2026-04-03
> >
> > Thanks for author's rebuttal. I have read other reviewers' comment and response.
> >
> > However, I still have some concerns.
> >
> > 1. Contributions:
> >
> >     - Some work has already done DIR-related work on LLM, such as [1,2]. I don't think simply shifting established problems of DIR [3] to MLLM is sufficient.
> >
> >     - I appreciate the author's analysis about limitations of SFT and RL, but such analysis has already appeared in other papers [4], and the author has not brought any new explanations or discoveries.
> >
> >     - I still thought the proposed method offers only incremental novelty built upon GRPO, which are also agreed by reviewer BUkJ "The level of originality appears moderate.", " the conceptual contribution remains incremental"
> >
> > 2. Beisdes, Reviewers Hgzh, rNoW, and I all expressed concern about the batch size issue. However, the newly additional experiment **clearly conflicts** with Table 7 in the manuscript, such as:
> >
> >     In additional results:
> >
> >     |Shot|All|Many|Med.|Few|All|Many|Med.|Few|
> >     |:-:|:-:|:-:|:-:|:-:|:-:|:-:|:-:|:-:|
> >     |24| 5.56±0.02 | 5.46±0.04 | 5.68±0.15 | 6.13±0.16 | 1.50±0.08 | 1.35±0.10 | 1.71±0.19 | 2.90±0.86
> >
> >     However, in original manuscript:
> >
> >     |Shot|All|Many|Med.|Few|All|Many|Med.|Few|
> >     |:-:|:-:|:-:|:-:|:-:|:-:|:-:|:-:|:-:|
> >     |24| 5.52 | 5.44 |  5.48 | 6.37 | 1.53 | 1.38 | 1.50 | 4.43
> >
> >     Given the discrepancies in the results, the reliability of the authors' experiment is questionable.
> >
> > Thus, I plan to maintain my score.
> >
> >
> > [1] IM-Context: In-Context Learning for Imbalanced Regression Tasks. TMLR 2024.
> >
> > [2] Are Foundation Models Useful for Bankruptcy Prediction? NeurIPS workshop 2025.
> >
> > [3] Delving into Deep Imbalanced Regression. ICML 2021.
> >
> > [4] Do llms understand user preferences? evaluating llms on user rating prediction. Arxiv 2023.

---

> > > ### Author Response · Authors · 2026-04-04
> > >
> > > ## On novelty and relation to prior imbalanced learning
> > > We thank the reviewer for pointing out these relevant works. We will include a more thorough discussion in the revised related work.
> > >
> > > Our contribution is driven by a **problem-first perspective**. We start from a key observation: numerical prediction in generative MLLMs differs from classical regression due to tokenized outputs, leading to a mismatch between token-level likelihood and value-level correctness. We systematically analyze SFT and standard GRPO, showing they fail to enforce consistent numerical structure under long-tailed distributions. **Our method is derived from this analysis, rather than a direct extension of prior work**.
> > > ### (1) IM-Context [1]: same goal, different paradigm
> > > IM-Context studies imbalanced regression via in-context learning (ICL), while we focus on **training-time optimization**. ICL performs inference via retrieval and context construction without parameter updates, operating on pre-extracted features (CLIP embeddings) and relying on carefully designed sampling.
> > >
> > > We directly optimize model parameters in generative MLLMs and introduce value-level, cross-sample supervision over complete numerical predictions. While ICL may capture numerical relationships within a context, such behavior depends on context design and does not generalize reliably across samples, as there is no mechanism to enforce consistent numerical structure at the model level.
> > > Empirically, our method **significantly outperforms** IM-Context: 5.52(3B) vs. 6.05 (AgeDB-DIR), and 6.71(3B) vs. 7.76 (IMDB-WIKI-DIR).
> > >
> > > ICL leverages existing model capability at inference time, whereas our method improves the model itself during training, leading to gains embedded in the model rather than dependent on context construction or sampling.
> > > ### (2) “Do LLMs Understand User Preferences?” [2]: different scope
> > > This work studies rating prediction under **pure SFT** with a small discrete output space (1–5, **single-token/digit**). It does not involve **multi-digit** prediction, long-tailed regression, or **RL-based training**, and therefore does not expose the tokenization-induced numerical inconsistency we study, nor analyze training limitations under such settings.
> > > ### (3) “Are Foundation Models Useful for Bankruptcy Prediction?” [3]: different task
> > > This work focuses on imbalanced **classification in a textual setting**, without numerical generation, multimodal inputs, or reward-based supervision, and is not directly comparable.
> > > ### (4) What is novel
> > > Our contribution is not simply applying GRPO with a new reward. Instead, we address a **previously underexplored limitation in MLLMs**: numerical values are generated as discrete tokens, breaking value-level consistency and making both SFT and standard RL insufficient under long-tailed distributions.
> > >
> > > We contribute:
> > > **Problem insight:** token-level vs. value-level mismatch; **Systematic analysis:** why SFT and GRPO fail; **New paradigm:** cross-sample, value-level RL supervision, shifting optimization from point-wise to distribution-level alignment. We also construct a **unified IQA benchmark** across four datasets (including IMDB-Movie and BoneAge).
> > >
> > > Our method results from systematic exploration across SFT and RL variants, not direct reuse of existing techniques.
> > > This is crucial in MLLMs, where training dynamics differ fundamentally from classical regression, making direct transfer of DIR methods **non-trivial** and substantially more challenging than implied.
> > >
> > > ### (5) Why this is not incremental
> > > Recent works (Visual-RFT, Rex-Omni, Perception-R1) show RL can improve core MLLM capabilities (e.g., classification, detection, grounding), even using relatively simple or task-derived rewards. Our work plays a role in advancing numerical regression as a core capability in MLLMs, providing a new supervision perspective.
> > > ### (6) External validation
> > > Reviewer BUkJ recognizes **novelty at the problem and perspective level**;
> > > Reviewer Hgzh emphasizes **importance of problem formulation and benchmark contribution**;
> > > Reviewer rNoW describes a **significant conceptual breakthrough** with strong empirical validation.
> > >
> > > We therefore suggest novelty should be assessed across **problem formulation, supervision paradigm, and empirical validation**—rather than reduced to method composition alone.
> > > ## On the reported discrepancies
> > > We thank the reviewer for raising this point. The discrepancy arises from different evaluation protocols.
> > >
> > > The main paper reports results with a fixed seed (42) for controlled comparison. Following the reviewer’s suggestion, we report multi-seed results (6 seeds). It is expected **multi-seed averages do not exactly match a single-seed result**.
> > >
> > > Importantly, standard deviations are small, indicating stable training and consistent performance, with trends unchanged across seeds.
> > > We will release full training and inference code upon acceptance to ensure reproducibility and further support reliability.

---

### Official Review · Reviewer_BUkJ · 2026-03-11

**Soundness:** 3
**Presentation:** 3
**Significance:** 3
**Originality:** 2
**Overall Recommendation:** 5
**Confidence:** 4

**Summary:**

This paper addresses the problem of numerical regression in multimodal LLMs under long-tailed target distributions, where standard token-level supervision and point-wise regression rewards lead to regression-to-the-mean behavior and poor tail performance. The authors observe that existing training paradigms fail to capture the global distributional structure of continuous targets, as predictions are optimized independently rather than in relation to other samples. To address this, they propose a distribution-aware reinforcement learning approach built on GRPO, where each sampled prediction is evaluated using a Concordance Correlation Coefficient (CCC)-based reward computed against a relational comparison vector that includes the mean predictions of other samples in the minibatch. This batch-level reward encourages alignment between predicted and ground-truth distributions in terms of correlation, scale, and mean, and can be integrated into existing GRPO pipelines without modifying the model architecture.

**Compliance With Llm Reviewing Policy:**

Affirmed.

**Final Justification:**

The rebuttal fully addressed my concerns. I maintain my positive recommendation.

**Key Questions For Authors:**

1. The proposed reward depends on minibatch-level statistics, which are inherently stochastic due to data shuffling and sampling. However, the reported results do not appear to include averages across multiple random seeds or variance measures. Could the authors clarify whether experiments were repeated across multiple seeds, and if so, provide the variance of the results? This would help assess the robustness and stability of the observed improvements.

2. Could the authors clarify whether any analysis has been conducted to assess the sensitivity of the CCC-based reward to minibatch composition beyond batch size? While this limitation is acknowledged in the appendix as a direction for future work, it is particularly relevant since the reward depends on relational comparisons within each batch. Additionally, as the results do not appear to be reported across multiple random seeds, it is unclear how robust the observed improvements are to stochastic variations in data shuffling and the RL training pipeline. Providing such analysis would help strengthen confidence in the stability and reproducibility of the proposed method.

3. How would the proposed method and especially the CCC-based reward generalize to the multi-dimensional regression setting ? Do you have any preliminary or exploratory findings for this case ?

**Limitations:**

yes

**Strengths And Weaknesses:**

**Soundness:** The submission is technically sound and the main claims are supported empirically rather than theoretically. The proposed CCC-based reward design is well motivated and aligns with the observed problem in numerical regression for MLLMs, and integrates naturally into the GRPO framework. The empirical evaluation includes relevant baselines and informative ablations, which help isolate the effect of the proposed method.

Weakness: The method relies on minibatch-level statistics as a proxy for global distributional structure, which may introduce noise when batches are small or not representative. While the authors provide empirical sensitivity studies with respect to batch size, the results are not reported across multiple random seeds. Given that minibatch composition depends on stochastic sampling, reporting seed-averaged results would help assess the robustness and stability of the proposed reward.

**Presentation:** The submission is clearly written and easy to follow. The method is described with sufficient detail, including the reward construction and integration into GRPO, the datasets construction, allowing an expert reader to reproduce the approach. The figures and tables are helpful in illustrating both the intuition and emprirical behaviour of the method. The related work section appropriately positions the contribution within prior work on deep imbalanced regression and RL-based post-training of MLLMs, and the distinctions from existing point-wise and architectural approaches are clearly explained.

**Significance:** The paper addresses an important limitation in MLLMs in performing numerical regression under long-tailed target distributions. This setting is common in real-world multimodal prediction tasks, yet remains underexplored in the context of generative models. The proposed approach provides a practical and general way to improve regression behavior without requiring architectural changes, making it relevant for existing post-training pipelines.

Weakness: The scope is focused only on one-dimensional regression tasks, although acknowleged by the authors, it can limit the generalization of the method and the claims in the paper.

**Originality:** The paper introduces a novel distribution-aware reward design for numerical regression in multimodal LLMs, framing the problem as distribution alignment rather than independent point-wise prediction. While the individual components are not new, their combination and application in this setting are original and well motivated. The work provides a meaningful extension to existing reward designs and introduces a useful perspective on improving regression behavior in generative models.

Weakness: The level of originality appears moderate. The approach primarily combines an existing reinforcement learning optimizer with a statistical reward function designed to capture distributional agreement. While this combination is reasonable and potentially useful in practice, the conceptual contribution remains incremental rather than introducing a fundamentally new learning paradigm

---

> ### Author Rebuttal · Authors · 2026-03-31
>
> ## Robustness to stochastic mini-batch composition
> We agree that robustness to stochastic mini-batch sampling is critical. We have now repeated our experiments across five random seeds and report mean$\pm$std in Table 1. The results are highly consistent. For example, on IMDB-Movie-DIR, our method achieves $6.89\pm0.07$ MAE and $2.17\pm0.08$ GM (3B), and $6.00\pm0.06$ MAE and $1.75\pm0.10$ GM (7B). Importantly, all seed runs consistently outperform SFT and point-wise RL baselines. This directly addresses the concern that gains may arise from stochastic batch composition rather than the reward design. We emphasize that we do *not* use any stratified or tail-aware sampling; mini-batches are formed via standard random shuffling. Stability arises from: (i) bounded reward ($\mathrm{CCC}\in[-1,1]$), (ii) bounded per-sample influence ($O(1/B)$), and (iii) GRPO stabilization (clipping + KL). Therefore, stochasticity introduces mild noise rather than unstable updates.
> |Metrics|MAE↓||||GM↓||||
> |---|---:|---:|---:|---:|---:|---:|---:|---:|
> |Method|All|Many|Med.|Few|All|Many|Med.|Few|
> |Ours-3B|**6.89±0.07**|**5.56±0.13**|**8.04±0.24**|**17.16±0.58**|**2.17±0.08**|**1.61±0.04**|**3.23±0.29**|**8.34±1.68**|
> |Ours-7B|**6.00±0.06**|**4.83±0.06**|**7.07±0.27**|**14.86±0.29**|**1.75±0.10**|**1.43±0.11**|**2.12±0.21**|**8.15±1.59**|
> ## Sensitivity beyond batch size
> We thank the reviewer for highlighting that sensitivity to *mini-batch composition* is distinct from batch size alone. Our analysis focuses on how changes in sample composition, rather than batch size itself, affect the CCC reward. Since the reward is computed
> from batch-level moments (mean, variance, covariance), its sensitivity is governed by how these statistics vary under resampling.
> First, under bounded targets, the influence of any *individual sample* on these batch moments is $O(1/B)$. More generally, moderate changes in batch composition induce bounded perturbations to the moments, and therefore bounded variation in the resulting CCC reward. Thus, composition changes introduce controlled stochastic perturbations rather than structural shifts in the supervision signal. Second, the CCC reward operates on *batch-level relational structure* (correlation, scale, and mean), rather than relying on particular sample pairs or the presence of rare tail instances. As a result, moderate variation in composition does not qualitatively change the supervision target, but only introduce statistical noise. Empirically, our multi-seed experiments capture stochastic variation from both data shuffling and RL sampling and show consistently low variance. Together with the batch-size ablations, these results provide supporting evidence that the method is not brittle to stochastic variation in mini-batch composition.
> We will clarify this point in the revision by explicitly distinguishing batch size from composition, and by emphasizing that the batch serves as a *moment estimator*, whose variability reflects statistical noise rather than instability of supervision.
> ## Extension to multi-dimensional regression
> We extend the framework to a 2D gaze estimation task (pitch and yaw). We apply the CCC reward per dimension and combine it with an angular consistency term. As shown in Table 2, the method consistently improves pitch MAE, yaw MAE, and angular error, especially in the few-shot regime with Qwen2.5-VL-3B. This demonstrates that our method remains effective beyond 1D scalar regression without architectural modification.
> |Metrics|PitchMAE↓||||YawMAE↓||||AngularError↓||||
> |---|---:|---:|---:|---:|---:|---:|---:|---:|---:|---:|---:|---:|
> |Shot|All|Many|Med.|Few|All|Many|Med.|Few|All|Many|Med.|Few|
> |PureRegression|3.54|3.06|3.42|4.30|7.19|5.67|7.31|9.03|8.54|6.94|8.54|10.59|
> |CCC|3.49|3.04|3.36|4.23|6.77|5.26|6.76|8.72|8.13|6.62|7.97|10.25|
> |CCCJoint|3.48|2.98|3.36|4.25|6.59|5.25|6.47|8.43|7.94|6.51|7.75|9.99|
> ## Originality
> We appreciate the reviewer's balanced assessment of originality. We would like to clarify that our contribution is not simply "GRPO + a statistical reward" in a conventional regression setting. The key challenge here is unique to *generative MLLMs*: numerical values are represented as sequences of discrete tokens, breaking numerical coherence. Classical DIR methods assume continuous outputs and cannot be directly applied. Our method instead operates on *complete numerical predictions* and introduces *cross-sample supervision* via a CCC-based reward. This shifts optimization from token-level likelihood to distribution-level alignment, which is not captured by existing DIR or RL formulations. While simple in form, we believe this addresses a previously underexplored limitation in MLLM numerical reasoning and introduces a new perspective on reward design for generative regression.

---

> > ### Author Rebuttal · Reviewer_BUkJ · 2026-04-02
> >
> > I thank the authors for the rebuttal and addressing my concerns about batch-level sensitivity, robustness across seeds and extension to multi-dimensional regression tasks, which strengthens confidence in the stability of the proposed method. Therefore I keep my positive assessement of the paper.

---

> > > ### Author Response · Authors · 2026-04-04
> > >
> > > Thank you very much for your thoughtful feedback and for your positive assessment of our work.
> > >
> > > We are glad that our clarifications regarding batch-level sensitivity, robustness across different random seeds, and the extension to multi-dimensional regression tasks have helped strengthen confidence in the stability of our method.
> > >
> > > We truly appreciate your support.

---

### Decision · Program_Chairs · 2026-04-30

**Decision:**

Accept (regular)

**Comment:**

This paper addresses the "regression-to-the-mean" phenomenon in Multimodal Large Language Models (MLLMs) when performing numerical regression on long-tailed distributions. The authors identify a fundamental "token-value mismatch," where token-level supervised fine-tuning (SFT) fails to preserve the continuous numerical topology required for accurate regression. To resolve this, they propose CCC-GRPO, a distribution-aware reinforcement learning framework that introduces cross-sample relational supervision. By utilizing a Concordance Correlation Coefficient (CCC)-based reward at the batch level, the model is incentivized to align predicted and ground-truth distributions in terms of correlation, scale, and mean.

The majority of the reviewer pool (BUkJ, Hgzh, rNoW) finds the work to be a significant conceptual shift from point-wise to relational supervision. While Reviewer ZzCn maintained a negative stance due to concerns over incremental novelty and experimental stability, the authors’ detailed technical responses and additional multi-seed data provide sufficient evidence of the method's robustness. The paper addresses a practical and underexplored limitation of MLLMs with a sound, well-validated framework. The authors should address reviewers'  comments in their final version.